



# The Greenland Ice-Marginal Lake Inventory Series from 2016 to 2023

Penelope How[1], Dorthe Petersen[2], Kristian K. Kjeldsen[1], Katrine Raundrup[3], Nanna B. Karlsson[1], Alexandra Messerli[2], Anja Rutishauser[1], Jonathan L. Carrivick[4], James M. Lea[5], Robert S. Fausto[1], Andreas P. Ahlstrøm[1], and Signe B. Andersen[1]

[1]Department of Glaciology and Climate, Geological Survey of Denmark and Greenland, Denmark
[2]Department of Hydrology and Climate, Asiaq Greenland Survey, Greenland
[3]Department of Environment and Mineral Resources, Greenland Institute of Natural Resources, Greenland
[4]School of Geography and water@leeds, University of Leeds, UK
[5]Department of Geography and Planning, University of Liverpool, UK

**Correspondence:** Penelope How (pho@geus.dk)

**Abstract.** The Greenland Ice Sheet and its surrounding peripheral glaciers and ice caps are projected to be the largest cryospheric contributor to sea level rise in the next century. While glacial meltwater is typically assumed to flow directly into the ocean, ice-marginal lakes temporarily store a portion of this runoff, influencing glacier dynamics, lacustrine-driven ablation, ecosystems, and downstream hydrology. The size, abundance and dynamics of ice-marginal lakes are expected to change in the
5 future. However, they remain under-represented in projections of sea level change and glacier mass loss. Here, we provide eight annual records across Greenland of lake abundance, lake surface extents, and surface water temperature estimates from 2016 to 2023. The dataset catalogs 2918 automatically classified ice-marginal lakes and reveals their evolving conditions over time. Our dataset fills critical gaps in understanding Greenland's terrestrial water storage and its implications for sea level change projections, providing a first step toward quantifying meltwater storage at ice margins. Equally important, it supports
assessments of ice sheet and glacier dynamics, such as lacustrine-driven ablation, and Arctic ecological studies of lake changes impacting ecosystems. The inventory series will also aid environmental management and hydropower planning aligned with Greenland's proposed commitments under the Paris Agreement. The inventory series is openly accessible on the GEUS Dataverse (https://doi.org/10.22008/FK2/MBKW9N) with full metadata, documentation, and a reproducible processing workflow (How et al., 2025).

## 1 Introduction

The Greenland Ice Sheet and its peripheral glaciers and ice caps (PGICs) are forecast to be the largest cryospheric contributor to sea level rise over the coming century (Arctic Monitoring and Assessment Programme (AMAP), 2021). At present, these projections assume that meltwater from the Greenland Ice Sheet flows directly into the ocean, yet a portion of this is known to be stored temporarily in ice-marginal lakes, along the ice edge of the Greenland Ice Sheet and in front of and beside
surrounding PGICs. The delayed release of meltwater at the ice margin is a significant, dynamic component of terrestrial storage. Ice-marginal lakes around the Greenland Ice Sheet form as meltwater becomes trapped at the terminus or edges of



an outlet glacier (How et al., 2021; Carrivick et al., 2022). Many of these lakes can be persistent and stable (Carrivick and Quincey, 2014), but an increasing number are recognised to be highly dynamic systems (Dømgaard et al., 2024). For example, many ice-marginal lakes in Greenland are prone to sudden and short-lived drainage events, thereby producing GLOFs (Glacial Lake Outburst Flood events) which are also referred to as *jökulhlaups* (Icelandic) or *sermimit supinerit* (direct translation into Kalaallisut, West Greenlandic; in singular *sermimit supineq*). GLOFs in Greenland can have characteristics of megafloods (Carrivick and Tweed, 2019) and have caused glacier speed-up events (e.g., Kjeldsen et al., 2017), influenced downstream erosion and sedimentation rates (e.g., Carrivick et al., 2013; Tomczyk et al., 2020), and water salinity (e.g., Grinsted et al., 2017; Kjeldsen et al., 2014).

The presence of an ice-marginal lake introduces a suite of thermo-mechanical processes, including lacustrine submarine melting and calving, that can dictate glacier margin morphology, dynamics and exacerbate ice mass loss (Sutherland et al., 2020; Carrivick and Tweed, 2019; Zhang et al., 2023b). At Russell Lake, situated beside Russell Glacier in Kangerlussuaq (West Greenland), these processes are relatively well-documented, illustrating how lake presence enhances melt-under-cutting and calving (Carrivick et al., 2017; Dømgaard et al., 2023). With continued retreat of the Greenland Ice Sheet under a warming climate, ice-marginal lakes and their associated processes are expected to become more abundant, larger, and warmer, which will likely amplify lacustrine-driven proglacial melt rates and GLOF events (Carrivick and Tweed, 2016; Grinsted et al., 2017; Shugar et al., 2020; Carrivick et al., 2022; Dye et al., 2021; Lützow et al., 2023; Rick et al., 2023; Veh et al., 2023; Holt et al., 2024; Zhang et al., 2024). However, ice-marginal lakes and their associated processes are largely absent from sea level change projections, which assume an immediate meltwater contribution to the ocean. This assumption overlooks the role of ice-marginal lakes as intermediary storage, and changes in lacustrine conditions, caused for instance when glaciers retreat onto land.

Mapping ice-marginal lakes is challenging due to the variability in lake characteristics. Remote sensing has been a viable approach for mapping the presence and surface extent of ice-marginal lakes, as demonstrated by inventories in Greenland (How et al., 2021), Alaska (Rick et al., 2022), Norway (Andreassen et al., 2022), Svalbard (Wieczorek et al., 2023) and High Mountain Asia (Chen et al., 2021). In general, classification approaches have been established to identify water bodies from SAR (Synthetic Aperture Radar) and multi-spectral (i.e. red, green, blue, near-infrared, shortwave) imagery, along with water potential identification using sink analysis from Digital Elevation Models (DEMs). As Greenland covers a large latitudinal range, ice-marginal lakes have very varying conditions which make them difficult to classify through one adopted method (How et al., 2021). For instance, surface sediment load, ice, and snow cover can vary significantly, with perennial ice cover in some cases at high latitudes and elevations (e.g., Mallalieu et al., 2021). Accordingly, multi-method classification approaches are necessary to capture this diversity (How et al., 2021).

Existing ice-marginal lake inventories are often static and therefore do not capture the dynamic nature of these lakes, nor capture new lakes and retire detached lakes once the margin has retreated. Given that ice-marginal lakes are projected to increase in size and abundance over time (Shugar et al., 2020; Zhang et al., 2024), it is of utmost importance to generate time-series that adequately capture ice-marginal lake change and assess the impact of these changes on future sea level projections.

Earth System
Open Access Science
Data Discussions

Here, we present an annual series of Greenland ice-marginal lakes from 2016 to 2023, classified using an established multi-method remote sensing approach. Each annual inventory maps the number of lakes (i.e. abundance) and lake surface area, along with attributes such as known lake name and surface water temperature estimations. These inventories reveal evolving lake conditions that support future assessments of sea level contribution, ecosystem productivity, and biological activity associated with the Greenland Ice Sheet and the PGICs.

## 2 Data description

### 2.1 Dataset overview

The annual inventory series is a follow-on from the 2017 Greenland ice-marginal lake inventory, largely adopting the same classification approach, data structure and formatting (How et al., 2021). The dataset consists of a series of annual inventories, mapping the presence and extent of ice-contact lakes across Greenland (Figure 1). Ice-contact lakes are defined as water bodies $> 0.05$ km$^2$ (based on satellite image spatial resolution), which are immediately adjacent to the Greenland Ice Sheet and/or the PGICs of Greenland. The annual inventory series spans the entirety of Greenland, including all terrestrial regions. Thus far, there are 8 annual inventories, covering 2016 to 2023, where one inventory represents one year.

### 2.2 Data sources and acquisition

Ice-marginal lakes are identified using three established classification methods, from Synthetic Aperture Radar (SAR) and multi-spectral imagery, and Digital Elevation Models (DEMs). Classifications from SAR and multi-spectral imagery for each inventory year are identified from all available Sentinel-1 and Sentinel-2 image acquisitions for the months of July and August (Table 1). DEM classifications are made from a static data product which covers the period 2008 to 2016. Metadata for each identified lake includes a lake surface temperature estimate, which is derived from Landsat 8/9 thermal band imagery (Table 1).

### 2.3 Data format and structure

The inventory series data are distributed as polygon vector features in an open GeoPackage format (.gpkg), with coordinates provided in the WGS NSIDC Sea Ice Polar Stereographic North (EPSG:3413) projected coordinate system. File names follow the convention defined in the original 2017 Greenland ice-marginal lake inventory (Wiesmann et al., 2021; How et al., 2021):

*<inventory-year>-<funder>-<project-acronym>-IML-f<version-number>.<file-extension>*.

Each GeoPackage file contains metadata regarding the lake description, physical measurements, lake surface temperature, method/s of classification, verification and possible editing (Table 2). A key piece of metadata to highlight is the lake identification number ("lake_id", Table 2), which are assigned to each classified ice-marginal lake, often consisting of multiple polygon features and/or classifications. These unique identifications are compatible across inventory years, therefore supporting time-series analysis and comparison across inventories.





**Table 1.** Summary of satellite data sources

| Satellite | Data product | Acquisition filters | Spatial resolution |
|---|---|---|---|
| Sentinel-1 | Ground Range Detected (GRD) dual-polarization C-band SAR images | Interferometric Wide Swath (IW), Horizontal-Horizontal (HH) polarisation, 01 Jul to 31 Aug | 10 metres |
| Sentinel-2 | Multispectral instrument (MSI), Top of Atmosphere (TOA), Level 1C images | 01 Jul to 31 Aug, 20% max. cloud cover | 10 metres |
| - | ArcticDEM mosaic (version 3) | - | 2 metres |
| Landsat 8/9 | Operational Land Imager/Thermal Infrared Sensor (OLI/TIRS), Collection 2, Level 2, surface temperature science product | 01 to 31 Aug, 30% max. cloud cover | 30 metres |

Information is included regarding whether the adjacent ice margin is either the ice sheet or PGIC ("margin", Table 2). This margin information originates from the MEaSUREs GIMP 15 m ice mask, previously used for the spatial filtering. In addition, each ice-marginal lake is assigned a region – north-west (NW), north (NO), north-east (NE), central-east (CE), south-east (SE), south-west (SW), and central-west (CW) ("region", Table 2). These regions and their corresponding names are based on ice

sheet catchment regions from Mouginot and Rignot (2019), which are used to also extend to the terrestrial periphery beyond the ice sheet. By doing so, regional trends can be identified from ice-marginal lakes with a PGIC margin as well as the ice sheet (Carrivick et al., 2022).

Lake names ("lake_name", Table 2) are assigned in instances where a name is available, with preference to West Greenlandic (Kalaallisut) placenames followed by Old Greenlandic and alternative foreign placenames. Placenames are provided

by Oqaasileriffik (the Language Secretariat of Greenland) placename database (https://nunataqqi.gl/), filtering placenames to those associated with lake features. The placename database is distributed with QGreenland v3.0 (Moon et al., 2023).

A readme file is included with the dataset that outlines the data file contents and terms of use. An additional data file is included which is a point vector GeoPackage file representing all identified lakes across the inventory series (presented in Figure 1). This includes manually identified lake locations that are not captured in the inventory series using the automated

classification approaches.





**Table 2.** Summary of metadata included with each ice-marginal lake inventory in the annual series

| Variable name | Description | Format |
|---|---|---|
| lake_id | Identifying number for each unique lake | Integer |
| lake_name | Lake placename, as defined by the placename database provided by Oqaasileriffik (the Language Secretariat of Greenland) (https://nunataqqi.gl/) which is distributed with QGreenland (https://qgreenland.org/). If no lake name is given then the placename is classed as "Unknown". | String |
| margin | Type of margin that the lake is adjacent to ("ICE_SHEET", "ICE_CAP") | String |
| region | Region that lake is located, as defined by Mouginot and Rignot (2019) ("NW", "NO", "NE", "CE", "SE", "SW", "CW") | String |
| area_sqkm | Areal extent of polygon/s in square kilometres | Float |
| length_km | Length of polygon/s perimeter in kilometres | Float |
| temp_aver | Average lake surface temperature estimate for the month of August (in degrees Celsius), derived from the Landsat 8/9 OLI/TIRS Collection 2 Level 2 surface temperature data product | Float |
| temp_min | Minimum pixel lake surface temperature estimate for the month of August (in degrees Celsius), derived from the Landsat 8/9 OLI/TIRS Collection 2 Level 2 surface temperature data product | Float |
| temp_max | Maximum pixel lake surface temperature estimate for the month of August (in degrees Celsius), derived from the Landsat 8/9 OLI/TIRS Collection 2 Level 2 surface temperature data product | Float |
| temp_stdev | Average lake surface temperature estimate standard deviation for the month of August, derived from the Landsat 8/9 OLI/TIRS Collection 2 Level 2 surface temperature data product | Float |
| temp_count | Number of Landsat 8/9 OLI/TIRS Collection 2 Level 2 scenes that lake surface temperature information were derived from. Scenes are only selected from the month of August | Integer |
| method | Method of classification ("DEM", "SAR", "VIS") | String |
| source | Image source of classification ("ARCTICDEM", "S1", "S2") | String |
| all_src | List of all sources that successfully classified the lake (i.e. all classifications with the same "lake_name" value) | String |
| num_src | Number of sources that successfully classified the lake ("1", "2", "3") | Integer |
| certainty | Certainty of classification, which is calculated from "all_src" as a score between "0" and "1" | Float |
| start_date | Start date for classification image filtering | String |
| end_date | End date for classification image filtering | String |
| verified | Flag to denote if the lake has been manually verified ("Yes", "No") | String |
| verif_by | Author of verification | String |
| edited | Flag to denote if polygon has been manually edited ("Yes", "No") | String |
| edited_by | Author of manual editing | String |



## 3 Methodology

### 3.1 Lake classification

Lake classifications are based on those adopted in the production of the 2017 Greenland ice-marginal lake inventory, which is summarised in Figure 2 (How et al., 2021). The main progression (and therefore difference) is that the processing pipeline is
now unified and operates through Google Earth Engine to conduct the heavy image processing (How, 2024), and filtering/post-processing conducted with open-source spatial packages in Python, namely geopandas (Kelsey et al., 2020) and rasterio (Gillies et al., 2013–). The Python pipeline is deployable as a package called GrIML (How, 2024), which is accompanied by thorough documentation and guidelines (https://griml.readthedocs.io). This ensures a high level of reproducibility and transparency that adheres to the FAIR (Findability, Accessibility, Interoperability, and Reusability) principles (Wilkinson et al., 2016).

#### 3.1.1 SAR backscatter classification

Water bodies are classified from Sentinel-1 GRD scenes, which are dual-polarization C-band SAR images (Table 1). Scenes are pre-processed using the Sentinel-1 Toolbox to generate calibrated, ortho-corrected data, specifically thermal noise removal, radiometric calibration and terrain correction using either the SRTM 30 or ASTER DEM. Scenes are then filtered to IW swath and HH polarisation, with image acquisitions limited to the summer months (1 July to 31 August) of each inventory year (2016
to 2023). Averaged mosaics for each year are derived from all summer scenes for each year of the inventory series at a 10 m spatial resolution. These mosaics are smoothed using a focal median of 50 metres. Classifications are derived from these averaged and smoothed mosaics using a static threshold trained for detecting open water bodies (How et al., 2021).

#### 3.1.2 Multi-spectral indices classification

Water bodies are classified from Sentinel-2 MSI, TOA, Level-1C scenes acquired for the summer months (1 July to 31 August)
of each inventory year (2016 to 2023) (Table 1). Clouds are masked using the cloud mask provided with each scene (QA60), masking out opaque and cirrus clouds. The bands of interest are extracted, specifically blue (B2), green (B3), red (B4), near-infrared (B8), and the two shortwave infrared bands (B11, B12) (Table 3). The shortwave infrared bands are resampled from 60 m to 10 m spatial resolution, and then averaged band mosaics are produced from all summer scenes for each inventory year.

Five spectral indices are used to classify open water bodies: 1) Normalised Difference Water Index (McFeeters, 1996); 2)
Modified Normalised Difference Water Index (Xu, 2006); 3) Automated Water Extraction Index (with shadow) (Feyisa et al., 2014); 4) Automated Water Extraction Index (no shadow) (Feyisa et al., 2014); 5) Snow brightness radio (How et al., 2021) (Table 3). The thresholds for the indices are chosen based on previous studies of ice-marginal lakes (How et al., 2021; Shugar et al., 2020), where positive classifications adhere to all thresholds.





**Table 3.** Summary of multi-spectral indices for ice-marginal lake classification from Sentinel-2 Level 1C scenes

| Spectral index | Equation | Threshold/s | Target |
|---|---|---|---|
| Normalised Difference Water Index (NDWI) | $(B3 - B8) \div (B3 + B8)$ | $< 0.3$ | Open water with shadowing |
| Modified Normalised Difference Index (MNDWI) | $(B3 - B11) \div (B3 + B11)$ | $> 0.1$ | Snow/ice in water |
| Automated Water Extraction Index (with shadow) (AWEIsh) | $B2 + 2.5 \times B3 - 1.5 \times (B8 + B11) - 0.25 \times B12$ | $> 2000 \ \& < 5000$ | Optimised sediment-loaded water without shadowing |
| Automated Water Extraction Index (no shadow) (AWEInsh) | $4 \times (B3 - B11) - (0.25 \times B8 + 2.75 \times B12)$ | $> 4000 \ \& < 6000$ | Sediment-loaded water with shadowing |
| Snow Brightness Ratio (BRIGHTNESS) | $(B4 + B3 + B2) \div 3$ | $< 5000$ | Snow-covered areas |

### 3.1.3 Sink classification

Water bodies are classified from the ArcticDEM 2-metre mosaic (version 3), which is compiled from the best quality Arctic-DEM strip files and manually adjusted to form a static data product (Table 1). The mosaic is smoothed using a focal median of 110 metres, and DEM depressions (i.e. where water pools) are filled over a 50-pixel moving window and subsequently subtracted from the original mosaic; producing the outline of a lake. It is noted that this is an indirect water classification method compared to the former two approaches (which directly detect water). Therefore, validation is required to confirm the presence

of water in classified DEM sinks, which will be elaborated further in the following subsection.

### 3.2 Summer surface water temperature estimation

A summer surface water temperature estimate is provided with each classified lake across inventory years. Surface water temperature estimates are derived from the Landsat 8 and Landsat 9 OLI/TIRS surface temperature data, which is a Collection 2, Level-2 science product that is part of a large Landsat re-processing effort (Table 1). Surface temperature estimates are gener-

ated from descending, day-lit Landsat 8/9 acquisitions with thermal infrared band information (30 metre spatial resolution) and auxiliary data (i.e. Top Of Atmosphere reflectance and brightness temperature), along with ASTER datasets (global emissivity and normalised difference vegetation index) and atmospheric data (geopotential height, specific humidity and air temperature) (Earth Resources Observation and Science (EROS) Center, 2020; Malakar et al., 2018).



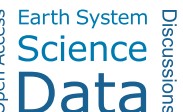

Due to the lack of in situ lake surface temperature measurements in Greenland, the scheme proposed by Dyba et al. (2022)
is adopted, whereby surface temperature values ($LST_{land}$) are corrected to surface water temperature ($LST_{water}$) using the
following calibration:

$$LST_{scaled} = LST_{land} \times 0.00341802 + 149.0 \tag{1}$$

$$LST_{water} = (0.806 \times LST_{scaled} + 54.37) - 273.15 \tag{2}$$

Where $LST_{scaled}$ is the applied scale factor for computing temperature in Kelvin (K) units, and $LST_{water}$ is the calibrated
surface water temperature in degrees Celsius (NASA Applied Remote Sensing Training (ARSET) program, 2022; Dyba et al.,
2022). This calibration has previously shown strong correlations against in situ measurements (average RMSE = 2.8 °C and $R^2$
= 0.8) from 38 lakes in Poland, highlighting accurate estimates through a simple linear calibration (Dyba et al., 2022). Ideally,
a correction factor specifically for calibrating values to Greenland lakes would be adopted. However, in situ validation datasets
in Greenland are sparse and the derived correction factor appears to agree well with the limited datasets available. In the future,
more in situ observations would strengthen the assessment, with a possibility to derive a Greenland specific correction scheme.

A summer average surface water temperature estimate is derived using this approach for each lake extent in the ice-marginal
lake inventory series. Scenes are filtered by a maximum cloud cover of 20%, with acquisitions limited to the month of August
to reduce the probability of ice-covered lake conditions. Lake extents are cropped by a border pixel (i.e. 30 metres) to reduce
the impact of edge effects. An average, maximum and minimum surface water temperature value is computed for each lake
extent over each inventory year, along with the standard deviation.

## 4 Results

### 4.1 Lake abundance

The dataset identifies 2918 automically delineated ice-marginal lakes (Figure 1). Of these lakes, 2054 share a margin with the
ice sheet whilst 864 are in contact with PGICs (Figure 3). The SW region holds the most lakes compared to other regions, with
786 classifications (640 classified at the ice sheet margin and 146 at the PGIC margins). A high abundance of PGIC lakes is
found in the NO region, with 278 lakes, compared to only 37 PGIC lakes in SE. This reflects the presence of more PGICs in
northern Greenland, compared to the greater ice sheet cover in the southeast.

Small fluctuations in the abundance of lakes are evident, fluctuating between 1963 (inventory year 2021) and 1827 (inventory
year 2022) lakes (Figure 3). The largest variation in lake abundance at the ice sheet is evident at the SW margin, with a
fluctuating range of 48 lakes (8%) between 567 (2018, 2023) and 615 lakes (2021) (Figure 3a). The CW and SE margin
experienced the least variability, only varying by 16 lakes and 15 lakes, respectively. Changes in lake abundance at the ice
sheet margin do not follow any spatial or temporal trends, with fluctuations unconnected to inventory year or margin region.

Lakes at the margins of periphery ice caps and glaciers vary between 723 (inventory year 2022) and 806 (inventory year
2019), with an average range of 11 lakes at the margins of periphery ice caps and glaciers (compared to an average range of
26 lakes at the ice sheet margin) (Figure 3b). The largest fluctuations in lake abundance are seen in the NO and NE regions,





fluctuating by 22 (8%) and 29 lakes (14%), respectively. This is linked to the higher number of lakes in these regions, which is supported by the smallest fluctuations evident in the regions where fewer lakes are present (i.e. NW, SE, CE and CW).

## 4.2 Lake surface extent

The largest lake in the inventory is Romer Sø, located in northeast Greenland, with a total area of 126.86 km$^2$ (Figure 1). The average lake size is 1.29 km$^2$, and the median lake size is 0.27 km$^2$ with 2395 lakes between 0.05-1.00 km$^2$ (82%). Only 59 lakes in the inventory series have a total area above 10 km$^2$ (2%). The NE and SW regions hold the largest lakes on average, with an average lake area of 1.63 km$^2$ (median: 0.34 km$^2$) and 1.58 km$^2$ (median: 0.27 km$^2$), respectively. On average, the largest PGIC lakes are also located in the NE region (1.43 km$^2$), likely because Romer Sø skews the PGIC average for this region.

The inventory series also holds information on the change in lake area over time, by comparing corresponding lake extents from each inventory year classified using a direct classification method (i.e. from SAR and/or multi-spectral imagery) (Figure 3). Change in average lake area over the ice sheet margin is relatively consistent across the inventory series, with the smallest change evident at the CE ice sheet margin (0.30 km$^2$) and the largest change evident at the NO ice sheet margin (1.31 km$^2$) (Figure 3a). Average lake size is highest in the NE region in 2022, with an average size of 2.77 km$^2$, coinciding with the lowest lake abundance (448). The average lake area is smaller across Greenland's PGIC margins, with an average lake area of 1.00 km$^2$, compared to lakes adjacent to the ice sheet with an average lake area of 1.40 km$^2$. Fluctuations in the average lake area across the inventory series are generally much smaller, apart from in the CE and NE regions which range across 2.20 km$^2$ and 1.76 km$^2$, respectively (Figure 3b).

Overall, lake area change trends can be tracked at 918 lakes in the inventory series (31% of all mapped lakes), with 243 experiencing growth between 2016 and 2023 (i.e. an increase in area of > 0.05 km$^2$), 675 declining in size (i.e. a decrease in area of > 0.05 km$^2$) and 778 remaining the same size (i.e. a change in lake area limited to ± 0.05 km$^2$) (Figure 4). The largest lake area changes are experienced at the larger lakes generally, such as those found in the NE region (Figure 4b) and the SW region (Figure 4d).

The inventory series demonstrates changes to lake morphology (and the corresponding change in ice margin morphology), of which four example scenarios are presented in Figure 5. A classic terminus basin retreat style is evident across many ice-marginal lake extents, as presented in Figure 5a, where terminus retreat/lake expansion is marked in the central section of the glacier outlet, leaving a trailing terminus morphology at the lateral margins. Peripheral terminus retreat is highlighted in Figure 5b, where terminus retreat/lake expansion is focused at the lateral margins. There are instances where the presence of a lake affects the boundary conditions of two glacier termini, as demonstrated in Figure 5c, where two glaciers terminate into the same common ice-marginal lake. And finally, there are instances displayed in the inventory series where there is margin retreat/lake expansion focused around a discrete zone, such as in Figure 5d where a marked embayment has formed at a particular point in the north region of the glacier terminus.



## 4.3 Lake surface temperature

An average surface temperature estimate is derived for each inventory lake from all available Landsat 8/9 scenes acquired in
the month of August for each inventory year (see Section 3.2). This information is provided in the metadata of the ice-marginal
lake inventory series. The average lake surface temperature fluctuates between 4.3 °C (2018) and 5.9 °C (2023) (Figure 6).
Fluctuations year on year vary, with instances of lake temperature falling between annual time steps (e.g. from 5.7 °C to 5.5
°C from 2019 to 2020), rising (e.g. from 5.5 °C to 5.9 °C from 2022 to 2023), and remaining consistent (e.g. between 5.4 °C
and 5.5 °C from 2020 to 2022).

## 5 Data quality and validation

### 5.1 Data quality control

Identified water bodies are compiled for each inventory year and filtered via three strategies: 1) by location; 2) by size; 3) by
manual curation (Figure 2). Firstly, lakes are filtered based on their location relative to the ice margin. Here, a 1 km buffer is
derived around the MEaSUREs GIMP 15 m ice mask and classified water bodies are retained if they are located within the
buffer (Howat, 2017; Howat et al., 2014). Classified water bodies are filtered by size, only retaining lakes above a minimum
size threshold of 0.05 km$^2$ based on the spatial resolution of the source satellite imagery; as adopted by How et al. (2021).
Finally, each inventory year dataset is manually curated to remove misclassifications, edit classifications (for example, where
the shadowing mask does not adequately remove shadowing effects), remove detected water bodies that do not hold water in
specific years, and remove water bodies that are detached from the ice margin. This manual curation is carried out via visual
inspection of Sentinel-2 TOA Level-1C true colour composites from each inventory year.

Classification information is provided with the ice-marginal lake inventory series, so that the performance of each classifi-
cation method can be evaluated (Figure 7). 14,020 of all detections in the inventory series (66%) are classified using only one
of the methods, composed largely from the DEM method (Figure 7a). 5156 detections are classified using two methods (24%),
and 2264 detections are classified using all three methods (11%). It is noted that the number of classification methods is not
a measure of certainty but instead should be interpreted as a reflection of lake appearance and its adherence to the criteria of
each classification method, as well as satellite data availability.

The SW region is typically where most lakes were classified with all three classification methods; across both the ice sheet
margin (Figure 7b) and the PGIC margins (Figure 7c). This is likely because the classification methods have been extensively
applied and developed in the SW region compared to others (e.g., Carrivick and Quincey, 2014; Carrivick et al., 2017; Kjeldsen
et al., 2017). The DEM method is heavily relied upon in the NO and NE regions where direct classification of open water is
challenging as lakes are more likely to be consistently ice/snow covered, and satellite image availability from Sentinel-1 and
Sentinel-2 can be limited (How et al., 2021).



## 5.2 Lake abundance error estimation

Previous error analysis suggested that the 2017 ice-marginal lake inventory captured 92% of lake abundance, based on com-
parison between the inventory and user-defined lakes over four regions at the NE, NW and SW ice sheet margin, and a region
within the PGICs, covering a collective area of 40,000 km². This formed an error estimate for lake abundance of ± 8% (see
How et al. (2021) for more details). As a follow-on to this effort, ice-marginal lakes are manually verified for each inventory
year, including those that were not classified using the automated methods. Across all inventory years, 4543 ice-marginal lakes
have been manually identified in total, of which 2915 (64%) are present in the ice-marginal lake inventory series. This forms a
revised lake abundance error estimation of ± 809 (36%). This error estimation is substantially different from the former esti-
mate because the 2017 ice-marginal lake inventory included manual lake delineations, whereas the inventory series presented
here only includes automated classifications (i.e. no manual lake delineations are included).

## 5.3 Surface lake temperature error estimation

Surface water temperature estimates are validated against all known and/or open-access in situ measurements of lake tem-
perature in Greenland (Figure 8). The only continuous/long-term in situ surface measurements (i.e. <= 2 m) are from six
lake records in southwest Greenland - Kangerluarsunnguup Tasia (64°07'50"N, 51°21'36"W) and Qassi-Sø (64°09'14"N,
51°18'27"W) (Greenland Ecosystem Monitoring, 2024), Russell Lake (67°13'77"N, 50°07'63"W) (courtesy of Kristian K.
Kjeldsen), and three lakes as part of the Asiaq Greenland Survey hydrological monitoring programme (Qassi-Sø 2024 mea-
surements; Qamanersuaq, 63°47'71"N, 50°00'50"W; and an unnamed lake referred to as Asiaq station 924, 64°12'99"N,
51°36'39"W).

Comparison of the 133 coinciding in situ measurements with those estimated using the remote sensing approach adopted
here exhibit a strong correlation (r² = 0.87), with an RMSE of 1.68 °C, suggesting that the remotely sensed temperature
estimates are reliable (Figure 8). This trend appears to be consistent regardless of the time of year. An error estimation of ±
1.2 °C is determined, based on the average difference from data points across all lake sites.

## 6 Potential applications and future updates

### 6.1 Uses for the ice-marginal lake inventory series

The inventory series presented here is the first step to quantify the terrestrial storage of meltwater, and how it changes over
time, which would be highly valuable for refining estimations of the future sea level contribution of the Greenland Ice Sheet
and surrounding PGICs. The ice-marginal lake inventory series is applicable to climate and cryosphere research, enabling
inter-annual comparison of lake change (abundance, extent and surface temperature) over time, similar to inventories for other
regions such as Svalbard (Wieczorek et al., 2023). Such inventories have been used to characterise ice dam types (e.g., Rick
et al., 2022), monitor GLOFs (e.g., Lützow et al., 2023), and assess lake conditions in catchments of interest (e.g., Hansen
et al., In Press). Lake conditions could also provide insights into glacier dynamics in lacustrine settings around Greenland, for





example, to investigate submarine melting in lacustrine settings and its impact on glacier retreat (e.g., Mallalieu et al., 2021).
More widely, the lake changes documented in this inventory series would be valuable to studies of the redistribution of mass on the earth surface, affecting gravity, geodesy and lithospheric elastic response (e.g., Ran et al., 2024).

Beyond scientific research, the inventory series will also be a useful resource in Greenland's assessment of infrastructure, with hydropower being the main sector that could benefit. Given Greenland's commitment to the Paris Agreement strongly suggests the expansion of current hydropower infrastructure, the ice-marginal lake inventory series could be valuable in infrastructure assessments (Naalakkersuisut, 2023). For example, the inventory series can be used to distinguish glacier-fed lakes from catchment-fed lakes, identify draining lakes, and other characteristics that are useful to discern viable catchment regions.

## 6.2 The future of the ice-marginal lake inventory series

It is planned to update the ice-marginal lake inventory series annually with new inventory years, using the methodology and data sources outlined here. A possibility could be to also include past years, prior to the Sentinel satellite era, however, this is limited by the open availability of SAR and multi-spectral satellite imagery at a high spatial resolution (i.e. 10 metres). Another avenue to explore is the inclusion of manually delineated lake extents where lakes have not been identified with the automated classification approaches. However, this would add further labour to the manual curation of the inventory series. An alternative would be to look at implementing new automated classification methods with machine learning, using the existing lake classifications as the foundation of a training dataset. Lake classification aided by machine learning has been successfully used for supraglacial lake detection on the Greenland Ice Sheet, so the use of machine learning in ice-marginal lake detection is likely to be feasible (Lutz et al., 2023; Melling et al., 2024).

One of the key limitations of this work to be addressed in the future is the reliance on static data products, in particular the static ArcticDEM 2 m mosaic for classification, and the MEaSUREs GIMP static ice margin for filtering. The use of static data products in the inventory series presented here highlights the importance of high-labour, time-consuming manual dataset curation. For the DEM classification, an alternative would be the ArcticDEM strip data product, which is time variant, but data coverage is lacking currently and scenes covering all Greenland are not possible from year to year. Another option would be to use coarser spatial resolution DEM products, such as PRODEM (500 m) (Winstrup et al., 2024), however, smaller lakes would not be identifiable. For the ice margin filtering, machine learning ice margin products show promise in being used in future editions of the inventory series, such as AutoTerm (trained with the TeamPicks dataset) (Goliber et al., 2022; Zhang et al., 2023a). The use of dynamic ice margin datasets in the future could negate the need for generating a classification spatial buffer around the margin data and instead classify ice-marginal lakes directly from their intersection with the ice margin position.

Another opportunity to further the inventory series would be the addition of valuable metadata on the characteristics and dynamics of each classified ice-marginal lake. The type of damming has been included in other inventories, proving to be useful for assessing present and future lake conditions under a changing climate (e.g., Rick et al., 2022). Incorporating known GLOFs and/or drainage periods for each lake would also provide insight into abrupt changes in terrestrial water storage and be highly valuable information for infrastructure assessments, such as hydropower utilities (e.g., Dømgaard et al., 2024).





## 7   Conclusions

Here, a series of annual inventories is presented that represent ice-marginal lake abundance, surface area extents, and surface temperature estimates across Greenland for the years 2016 to 2023. Ice-marginal lakes are mapped across the margin of the
Greenland Ice Sheet and its surrounding PGICs. The dataset demonstrates lake change over the 8-year period, which can be assessed at various scales, from individual lake, to regional, to Greenland-wide change. The annual ice-marginal lake inventory series is openly available on the GEUS Dataverse with a cite-able DOI at https://doi.org/10.22008/FK2/MBKW9N, including supporting metadata and documentation (How et al., 2025).

With each year, a new addition will be added to this dataset, with the hope that the inventory series could be used in the future
to assess lake change at multi-decadal time scales. This is supported by GrIML, an open processing workflow with open-source programming that is accessible to novice programmers with thorough documentation and straightforward deployment (How, 2024).

The annual ice-marginal lake inventory series is a valuable addition to addressing current limitations in terrestrial water storage and its influence on Greenland's future sea level contribution. This dataset is the first step towards quantifying meltwater
storage at the margins of the Greenland Ice Sheet, and surrounding PGICs. It also provides insight into lake change over time, and the resulting impact on glacier dynamics, such as lacustrine frontal ablation (i.e. submarine melting and calving). Beyond the cryospheric science community, the dataset will be invaluable to related disciplines in biology and ecology, where changes in lake conditions shape Arctic ecosystems and biological activity. On a national level, the inventory series could be a useful resource in environmental management and infrastructure assessment, for instance in the expansion of hydropower utilities as
suggested in Greenland's new commitments to the Paris Agreement.

## 8   Code and data availability

The dataset is openly available on the GEUS Dataverse at https://doi.org/10.22008/FK2/MBKW9N (How et al., 2025), distributed under a CC BY 4.0 license (https://creativecommons.org/licenses/by/4.0/). If the dataset is presented or used to support results of any kind then we ask that a reference to the dataset be included in publications, along with any relevant publications
from the data production team. If the dataset is crucial to the main findings, we encourage users to reach out to the authorship team as this will likely improve the quality of the work that uses this product. The production code for making the inventory series is openly available at https://github.com/GEUS-Glaciology-and-Climate/GrIML (How, 2024). It is distributed as a deployable and version-controlled Python package. If the production code is used or adapted, then we ask for a reference to be included in publications.

*Author contributions.*   P.H. led the production workflow and dataset presented, with input from D.P., K.K.K., N.B.K., A.M., A.R., J.L.C. and J.M.L. Validation datasets were collected and curated by D.P., K.K.K. and K.R. Management of the project and work presented was overseen by R.S.F., A.P.A. and S.B.A. All authors contributed to the manuscript text.





*Competing interests.* The authors declare that there are no competing interests.

*Acknowledgements.* P.H. was supported by an ESA (European Space Agency) Living Planet Fellowship (4000136382/21/I-DT-lr) entitled
"Examining Greenland's Ice Marginal Lakes under a Changing Climate (GrIML)". Further support was provided by the Programme for
Monitoring of the Greenland Ice Sheet (PROMICE), funded by the Geological Survey of Denmark and Greenland (GEUS) and the Danish
Ministry of Climate, Energy and Utilities under the Danish Cooperation for Environment in the Arctic (DANCEA), conducted in collab-
oration with DTU Space (Technical University of Denmark) and Asiaq Greenland Survey. In situ lake temperature datasets are supported
by the GrIML project (with measurement collection led by Asiaq Greenland Survey), BioBasis under the Greenland Ecosystem Monitoring
Programme (GEM), and the Greenland Integrated Observing System (GIOS) under the Danish Agency for Higher Education and Science.
K.K.K. acknowledges support from the Independent Research Fund in Denmark (grant ID 10.46540/3103-00234B). J.M.L. acknowledges
support from his UK Research and Innovation (UKRI) Future Leaders Fellowship (MR/X02346X/1). Additional thanks to Stephen Plummer
and Marcus Engdahl for technical advice and support, and Sikkersoq Olsen and Arnaq Brandt Johansen from Oqaasileriffik (the Language
Secretariat of Greenland) for clarification on the Kalaallisut terminology for GLOFs.





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



**Figure 1.** An overview of the abundance of lakes in ice-marginal lake inventory series, 2016-2023. Each mapped point denotes a unique lake, mapped across the Greenland Ice Sheet margin (blue) and the surrounding PGIC margins (white). The tables associated with each region present general statistics, with red starred points on the map corresponding to the largest lake of each region. Placenames for the largest lakes are sourced from the placename database provided by Oqaasileriffik (the Language Secretariat of Greenland), with inventory identification numbers presented where a name is not given. It is noted that the name of the largest lake in the CE region (Catalina Lake) is not present in the placename database, and instead we adopt the lake name from Grinsted et al. (2017). The catchment regions are those defined by Mouginot and Rignot (2019). Base maps for plotting are from QGreenland v3.0 (Moon et al., 2023).





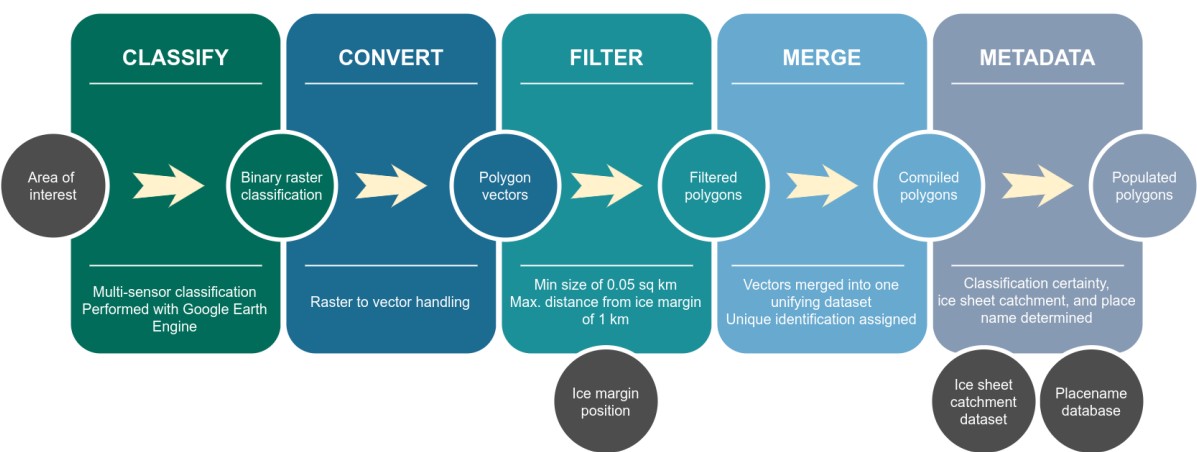

**Figure 2.** A visualisation of the processing workflow for the generation of the ice-marginal lake inventory series, including components performed with © Google Earth Engine ("Classify") and the Python package GrIML (How, 2024), which utilises Python spatial data handling packages geopandas (Kelsey et al., 2020) and rasterio (Gillies et al., 2013–). The workflow is based on How et al. (2021). The annotated rectangles refer to process stages (reading from left to right), the coloured annotated circles represent intermediary outputs to the corresponding process stages in the workflow, and the grey annotated circles represent workflow inputs.





**Figure 3.** Change in the abundance and average area (km$^2$) of ice-marginal lakes around the ice sheet margin (a) and PGIC margins (b). Each of the coloured bars denote lake abundance per region for a given year of the inventory series (2016-2023), with annotated numbers corresponding to the number of lakes classified for each region. Each line plot indicates the average lake area per region for a given year of the inventory series. Average lake area is compiled from all lakes classified from SAR and multi-spectral imagery, as DEM classifications are not a direct detection of water bodies.

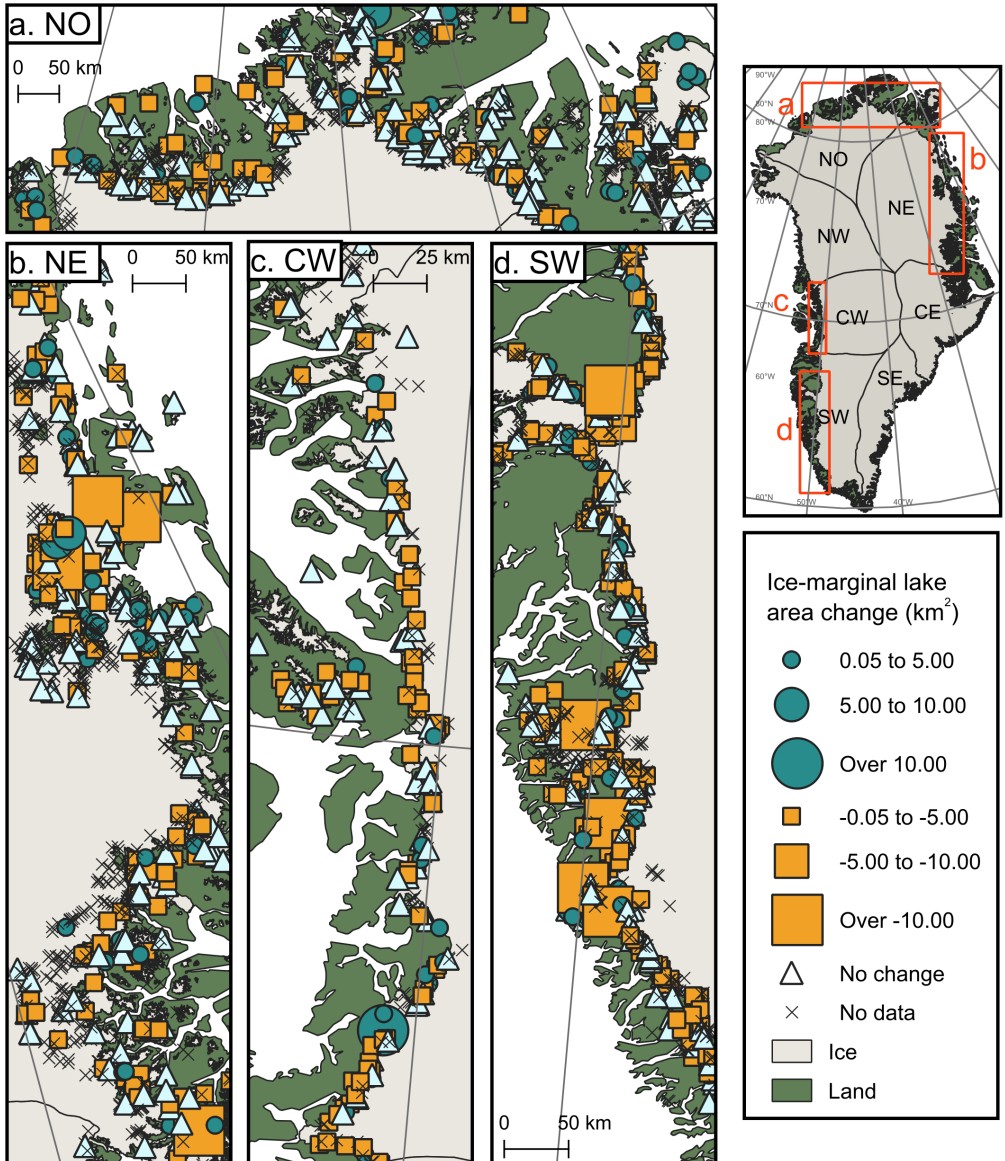

**Figure 4.** Change in lake area across the ice-marginal lake inventory series, 2016-2023. Example regions are highlighted from NO (a), NE (b), CW (c), and SW (d) regions of both the ice sheet and the PGICs. Lake area increase (purple circles), lake area decrease (yellow squares), and unchanged lake areas (white triangles) are mapped, with the size of the symbol denoting the amplitude of change (km$^2$). Each point denotes the change in lake size across the inventory series, as classified using the SAR and multi-spectral imagery methods. Lakes with no available area data (i.e. not classified using the SAR and multi-spectral imagery methods) are marked with crosshairs. The catchment regions are those defined by Mouginot and Rignot (2019). Base maps for plotting are from QGreenland v3.0 (Moon et al., 2023).

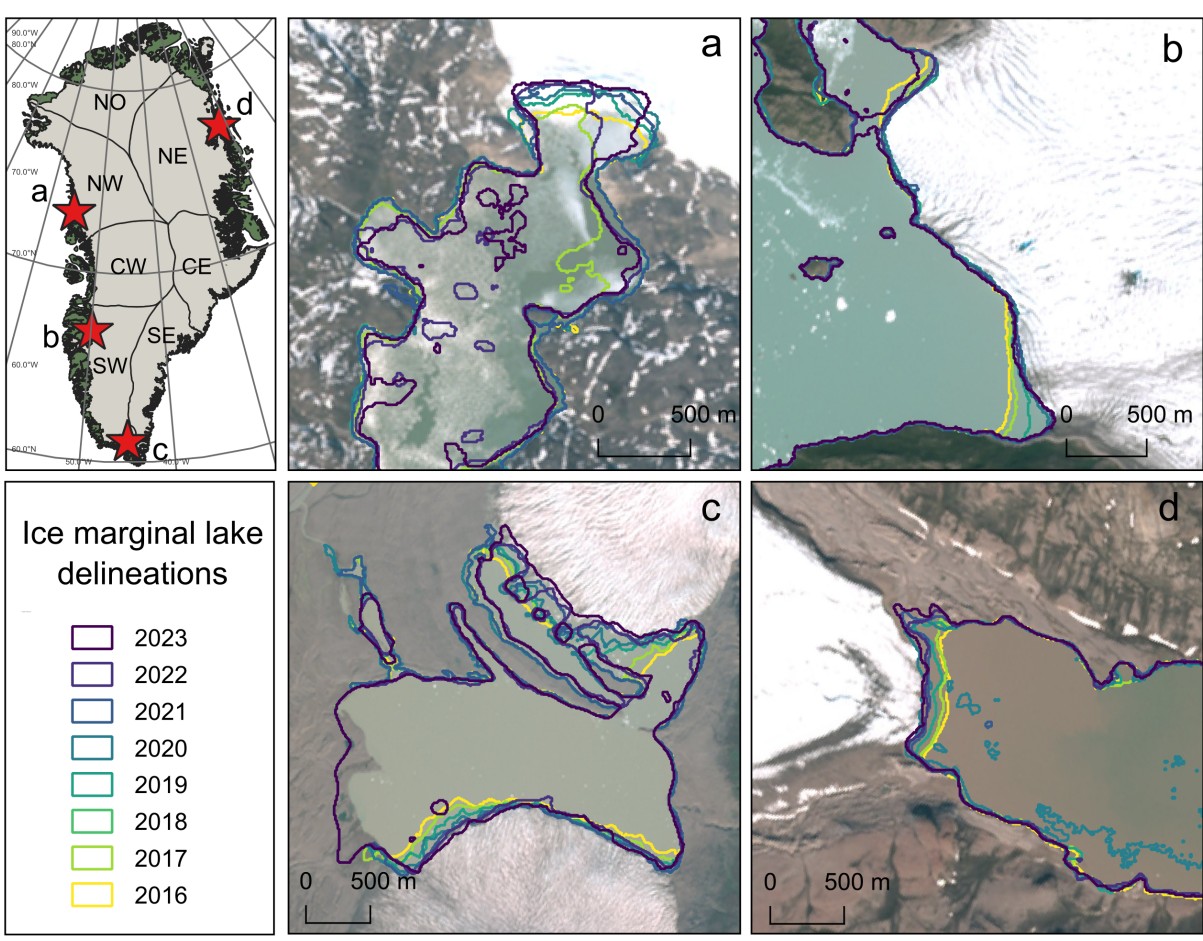

**Figure 5.** Examples of lake morphology change, and the corresponding evolution of ice termini morphology, from the ice-marginal lake inventory series. These examples highlight basin margin retreat (a), peripheral margin retreat (b), bilateral margin retreat (c), and focused margin retreat (d). The background satellite imagery presented is from a Sentinel-2 10 m 2022 mosaic (Styrelsen for Dataforsyning og Infrastruktur, 2024). The base layers for the insert map plotting are from QGreenland v3.0 (Moon et al., 2023).



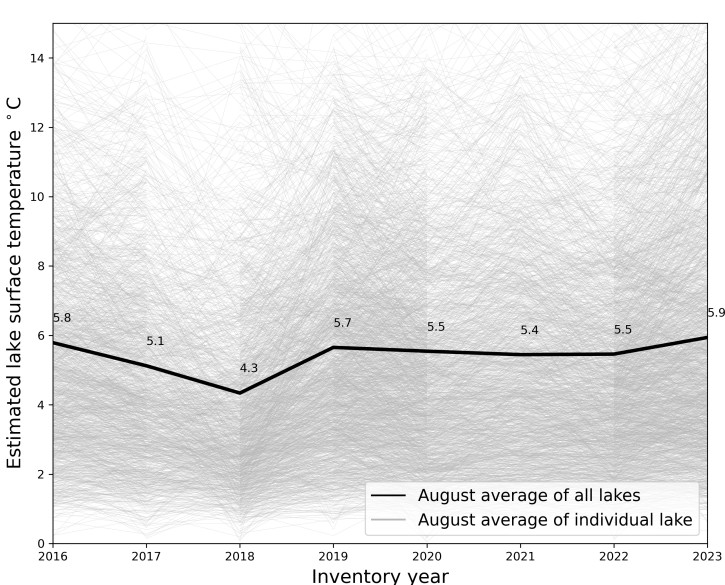

**Figure 6.** Average surface lake temperature estimates from the month of August at each inventory lake for each inventory year (2016-2023) (grey), with the average of all lakes overlaid (black). Surface lake temperature is derived from Landsat 8 and Landsat 9 OLI/TIRS Collection 2 Level 2 surface temperature data product. Averages are calculated from all available scenes acquired from the month of August to limit the risk of mis-estimates due to ice-covered conditions.

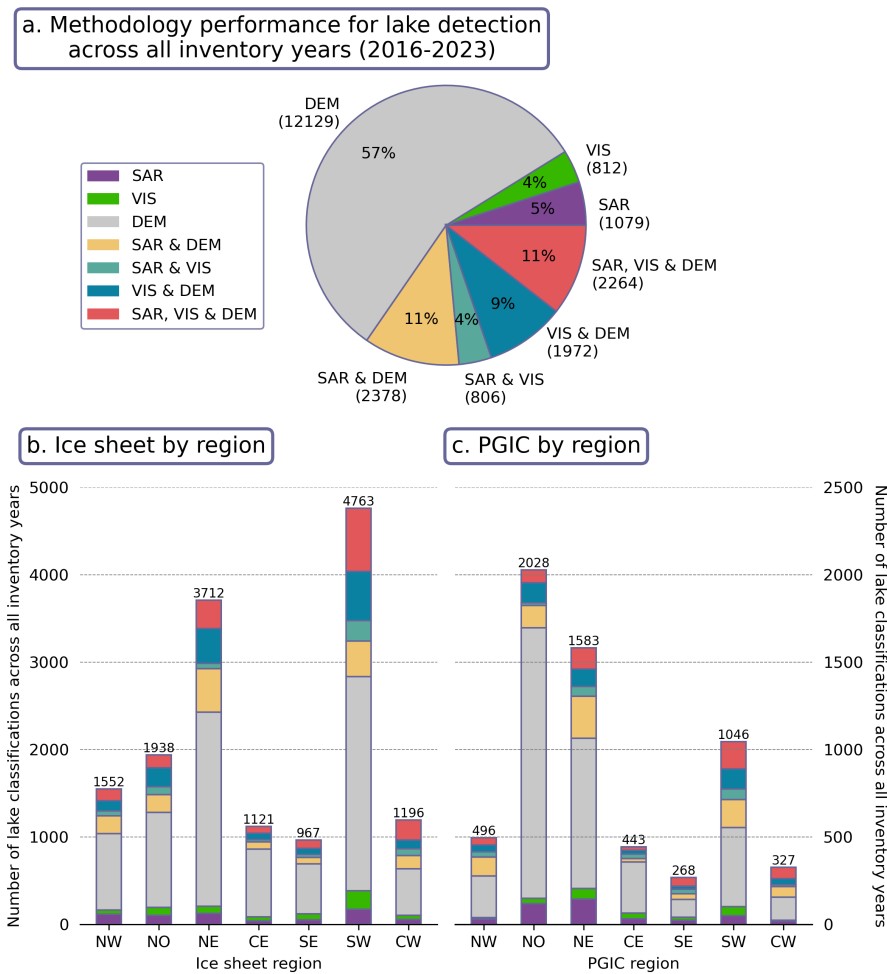

**Figure 7.** Lake classifications by method across the ice-marginal lake inventory series, with an overview of lake classifications over all inventory years (a) and classifications by region for lakes adjacent to the ice sheet (b) and the PGICs (c). SAR refers to the SAR backscatter classification method from Sentinel-1 imagery, VIS refers to the multi-spectral classification method from Sentinel-2 imagery, DEM refers to the DEM sink detection method from the ArcticDEM, and listed methods refer to instances where more than one method has been used to classify a lake (e.g. "SAR & DEM", "SAR, VIS & DEM"). The legend and colour scheme in (a) correspond to (b) and (c). Values in brackets in (a) are the absolute number of lakes corresponding to the provided percentages. The values printed on top of the bars in (b) are the total number of classifications in the given region.



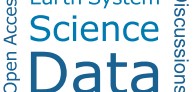

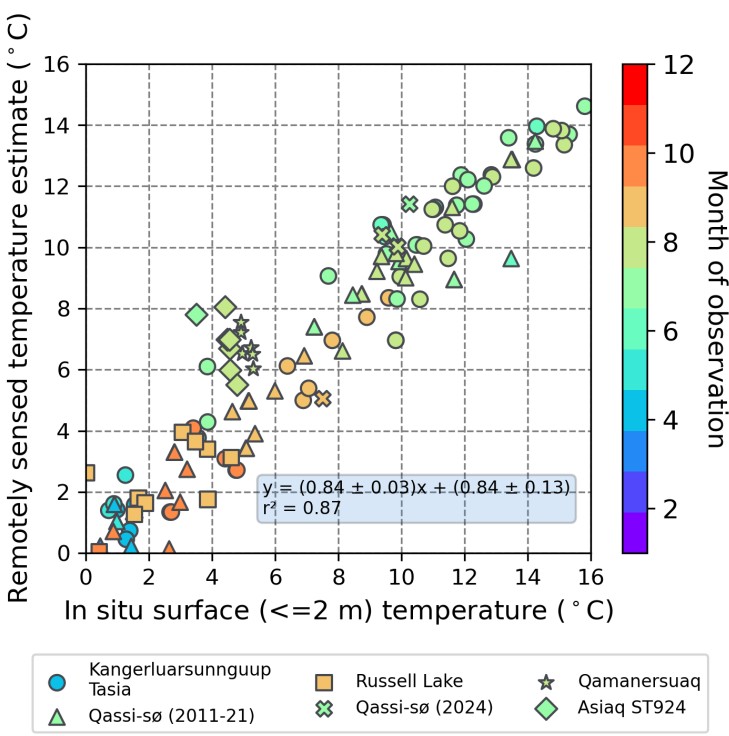

**Figure 8.** Comparison of in situ surface (<= 2 m) water temperature measurements with remotely sensed temperature estimates (°C) from Kangerluarsunnguup Tasia (circle), Qassi-Sø (2011-21) (triangle), Russell Lake (square), Qassi-Sø (2024) (cross), Qamanersuaq (star) and Asiaq station 924 (ST924) (diamond). The colour of each point corresponds to the month that the observation was collected.