# Peer review of "The Greenland Ice-Marginal Lake Inventory Series from 2016 to 2023"

_Earth System Science Data, 2025_

## Author Comment (AC1)

**Response to Reviewer #1**

This paper presents an annual ice-marginal lake inventory from 2016 to 2023, classified using an established remote sensing approach. The paper is short. Figures and Tables are nicely drafted. The dataset is certainly valuable, but the authors also report that their automated processing are underdetecting the lakes but apparently do not take any means to improve it. I miss more information on validation and more discussion on how improvements could be made. I also question why there are so many authors (I counted 12 incl 3 'managers') on this rather short paper not involving any field investigations.

Thank you for your feedback. This publication and the associated dataset are the culmination of a long-term effort across different projects, incorporating in situ data (and thus fieldwork) that has been collected by many contributors. We greatly appreciate your feedback and will respond to your major comments subsequently, followed by outlining our responses to your minor comments.

We appreciate your feedback on the automated classification method. The automated classification approach prioritizes consistency and reproducibility across large spatial and temporal scales. Incorporating manual delineations, while potentially improving detection in localized cases, would introduce subjectivity and limit the reproducibility and scalability of the dataset. That said, we recognize the potential for improving the classification methods in future work, including exploring hybrid approaches or enhanced algorithms that could reduce omission errors while maintaining automation and consistency. In Section 6.2 (The future of the ice-marginal lake inventory series), we have now elaborated on how the automated classification method could be improved upon with examples from other publications, and how the overall product could be developed to address the under-detection of lakes.

With regards to the validation, we had included error analysis for lake abundance and surface lake temperature, and data quality control as reflected in the methodology performance at the time of submission. In light of your feedback, we have now included an extra section that performs error analysis for lake size (this was also requested by Reviewer #2), therefore providing a more complete error analysis of each aspect of the dataset and the underlying methods. This error analysis can be found in Section 5.3 (Lake size error estimation). The discussion and other relevant sections have also been updated accordingly.

**Minor comments**

I have checked the readme.txt, and downloaded one of the lake files. Seems fine

**Great to hear. Thank you for checking the dataset.**

7 'The dataset catalogs 2918 automatically classified ice-marginal lakes and reveals their evolving conditions over time.' Does not the number of lakes vary through time?

Yes, the number of lakes varies through time as elaborated on in Section 4.1 (Lake abundance). The sentence is merely providing an overview of the number of unique lakes identified across all years. To make this clearer, we have edited this sentence: "Here, we present an eight-year (2016–2023) inventory of 2918 automatically classified ice-marginal lakes (>=0.05 km2) across Greenland, tracking changes in lake abundance, surface extent, and summer surface temperature over time." (Line 5-7)

**26 do you have a reference for this sentence?**

There are many broad references we could add here related to the study (and naming) of GLOFs/Jökulhlaups, as they are well-studied phenomena. References to examples where the terminology are used (Taylor et al., 2023; Elbi et al., 2023) have been added accordingly. With regards to a reference for the Greenlandic naming, this is based on direct consultation with Oqaasleriffik (the Language Secretariat of Greenland) as we had not come across a Greenlandic term for this previously. This information has now been added:

"For example, many ice-marginal lakes are prone to sudden and short-lived drainage events, thereby producing GLOFs (Glacial Lake Outburst Flood events) (e.g., Taylor et al., 2023) which are also referred to as jökulhlaups (Icelandic) (e.g., Eibl et al., 2023) or sermimit supinerit (direct translation into Kalaallisut, West Greenlandic; in singular sermimit supineq) (Oqaasileriffik, personal communication, November 2024)." (Line 28-31).

30-41 I found this a bit detailed, is all needed? I suggest shortening this part, do it a bit wider in terms of authors cited and more general before zooming in on Greenland.

We have removed the Russell Lake example to present a more general picture of icemarginal lakes. The paragraph is more concise now. 55 I am not sure that you in your paper defend the statement 'and assess the impact of these changes on future sea level projections.' There is no reference for the statement and I would be careful with it.

Agreed. We have changed this statement to address this concern: "Given that ice-marginal lakes are projected to increase in size and abundance over time (Shugar et al., 2020; Zhang et al., 2024), it is of utmost importance to generate time-series that adequately capture ice-marginal lake change and could potentially contribute to future sea level assessments." (Line 56-58).

63-65 here you use both ice-marginal and ice-contact lakes- you define ice-contact lakes, but not the other. I suggest you define both and check the use throughout.

Noted. We decided to drop the use of the term "ice-contact lake" and only use "ice-marginal lake". This has been checked and revised throughout the manuscript.

68./103/ can you here or elsewhere mention if there are any previous lake inventories based on Landsat that you use for reference or if this is the first?

Carrivick and Quincey (2014) presented an ice-marginal lake inventory using Landsat image classification which was multi-temporal, however, this was limited to the southwest Ice Sheet margin. How et al. (2021) provided the first Greenland-wide inventory, but this was a static dataset. Therefore, as far as we are aware, this is the first multi-temporal icemarginal lake inventory series that covers all of Greenland's Ice Sheet and PGIC margins. The inventory series presented builds upon the static 2017 inventory presented in How et al. (2021), which is already noted in this section (Line 65-66). We have now added this additional information on Lines 66-68: "How et al. (2021) provided the first Greenlandwide ice-marginal lake inventory as a static dataset, building upon regional multi-temporal efforts, such as the southwest inventory classified from Landsat imagery (Carrivick and Quincey, 2014)."

Can you specify that 2016 is the first year due to launch of Sentinel. It is implicit but not explicitly stated.

Done - "Thus far, there are 8 annual inventories, covering the Sentinel satellite era from 2016 to 2023, where one inventory represents one year." (Line 72-73)

**105 remove -?**

The Gillies et al. reference has a dash included at the end of the date, as specified by the authors (see here - https://github.com/rasterio/rasterio/blob/main/CITATION.txt). This is often the case when referencing software, to signify that updates and development of the software remain ongoing. We respect the wishes of the software authors and will not alter the reference as part of this review.

3.1.1./3.1.2./3.1.3/3.2/5.1/throughout where your work are described. Usually work done in methods are written using past tense. If you did the work, use past tense. This makes it easier differencing published work (present tense) from what has been done for this paper/dataset.'

Done.

163+ will this number of lakes sharing margin not differ with time, this can change for year to year, which year of the dataset are you referring to? Suggest to rewrite this. You mention below that lake number vary from year to year, so how come one number is static while other differs.

As previously explained, this statement is merely providing an overview of the number of unique lakes identified across all years. We have re-worded the statement to better convey this: "In total, the dataset identifies 2918 automatically delineated ice-marginal lakes across all inventory years". (Line 172).

209 past tense, was?

Done.

243 can you elaborate a bit more on how this was done and explain this better, it is a huge difference. Were lakes then included annually based on this effort? What was the follow up from the results you found. Did you try to improve the mapping method or include manual digitisation? Not clear to me. The last sentence '...no manual lake delineations are included' seems to be a strange follow up if the underdetection is so substantial. I am not sure if any of the statements in 6.1 is valid if the dataset is missing so many lakes. Could the methods have been improved? Or could manual updated help?

See major comments.

The result of the validation is completely missing from conclusion and abstract, data uncertainty and accuracy is important part of an ESSD paper. I would rewrite the abstract to be less general and more direct on results and uncertainties. same with conclusion.

Done. The Conclusions and Abstract have been written to focus more on the dataset results and validation.

I miss a figure showing validation of the method from one of the newer years that is previously not published.

We are unsure exactly what kind of figure is being requested here. Given a new validation section has been added to the manuscript (Section 5.3 Lake size error estimation), we have decided to take no further action on this comment.

**Response to Reviewer #2**

The manuscript provides an overview of the multi-temporal high resolution (spatial and thermal) development of ice marginal lake inventories of Greenland from 2016 to 2023. This represents a substantial advance in the science of ice marginal lake evolution at the regional scale across very remote. However, I do have some concerns over some of the parameters and lack of boundary error estimation within the manuscript. These could all be relatively easily address to provide a high quality inventory that will provide a substantial advancement in the study of ice-marginal lakes relationship with climate change in order to understand their role as a resource but also a hazard in some areas.

The combination of the multi-sensor remote sensing approach for lake detection following How et al., (2021) provides a relatively robust approach for a complex dynamic problem over a very large and mostly very remote area. The accurate detection of water bodies at these locations are not only important glaciologically, but also for hydrology and ecology downstream. With this in mind I would highly recommend that lakes are not eliminated from the inventory once they lose contact with the ice front, but instead are classed as 'non-contact' or perhaps archived in a sub-inventory. The presence of these lakes will further modify any glacial meltwater, as they can be a substantial sediment sink (Vowels et al., 2025) as well as thermal modification and also implications for passage of GLOFs. All of which have substantial importance downstream.

The detection of water bodies at these ice marginal lake sites across Greenland is an unenviable task – given the presence of snow and ice as well as frequent cloud cover. The

detection of 'lake margins' becomes a critical problem in analysing the spatial evolution. Some of these lake margins will have substantial snow and ice banks throughout most years. Consequently uncertainty assessment of lake margin boundaries becomes complicated, but still important – yet this does not appear in the manuscript? The combined areal uncertainty from How et al., (2021) could be referred to. This could be dealt with through standard remote sensing approaches to lake boundary errors or there is potential for classifying or flagging lake margins that have substantial snow/ice cover. This will effect the thermal conditions in the lake. It is also important for understanding some of the large lake decreases that your report – are these from increased snow/ice cover? Drainage/lowering? Or glacier terminus 'advance' from increased glacier velocities? If the latter then they are very important to examine further (King et al., 2018).

Unfortunately the Landsat Level 2 thermal product is resampled to 30m, which creates problems for eliminating the original 100m pixels that would be a combination of water and non-water. As well as the added uncertainty with defining lake boundaries and also any uncertainty with the alignment/correlation between the Landsat thermal sensor and infrared/visible sensors. In order to reduce pixels with thermal contamination in the dataset from boundary issues I would strongly recommend setting a buffer of 100m minimum around the lake margins. I think this will reduce some of the noise in the dataset, particularly for smaller lakes and those with peninsulas/rock islands etc.

This is a huge body of work that will have a transformative impact on glacial lake science (after a few modifications)!

Thank you for your feedback, Adrian. The thought and time taken to provide this detailed review is greatly appreciated. Responses to your major comments are provided below, followed by comments from your line-by-line feedback.

Based on your feedback, the dataset has now been updated with:

- 1. Centroid positions added as an attribute ("*centroid*"). These are XY central coordinates for each lake, derived from all classifications across the inventory series. Coordinates are provided in the WGS NSIDC Sea Ice Polar Stereographic North (EPSG:3413) projected coordinate system
- Landsat image acquisition periods have been added (attribute: "temp\_date"), with acquisition datetimes formatted as "YYYY-MM-DD HH:MM". In the case that averages are composed of multiple Landsat images, each datetime is listed with a ", " break
- 3. A new data file has been added which holds all classified ice-marginal lakes across the inventory series (*"ALL-ESA-GRIML-IML-<version>.gpkg"*). This version is an

aggregate of all polygons, merged based on corresponding lake identification numbers. In other words, one polygon vector feature signifies the maximum extent of one classified lake.

The revised dataset can be found at https://doi.org/10.22008/FK2/MBKW9N (version 2).

We have decided not to include ice-contact and non-contact water bodies (and a corresponding classification criterion in the metadata) primarily because this changes the inherent definition of the dataset and would be a substantial undertaking that would culminate in the production of a different dataset. Ice-marginal lakes are defined specifically as water bodies that share a boundary with an ice margin, and their presence has a direct impact on ice margin conditions. The term "glacial lake" is a broader term for ice-contact and non-contact lakes, including those that once shared a boundary with an ice margin (i.e. "detached") (Shugar et al., 2020). Broadening the dataset to include all glacial lakes would throw into question exactly what can be defined as a glacial lake, given that all lakes in Greenland were at some point connected to the Ice Sheet or PGIC and therefore could be defined as a glacial lake. In addition, it is expected that a high majority of noncontact lakes are stable and therefore there is no inherent need for an annual time-series of lake change in these cases, compared to ice-marginal lakes which are more changeable and dynamic. We believe the ice-marginal lake inventory series could form a component of a future glacial lake inventory though, which would be a separate dataset to the inventory series presented here. We have therefore included this idea as a future opportunity in Section 6.1 (Uses for the ice-marginal lake inventory series) (Line 301-304).

With regards to the issues with classified lake boundaries, an uncertainty assessment and error estimation is now included for the lake boundary/size. Reviewer #1 made a similar comment regarding the validation also. We have added this to the manuscript in Section 5.3 (Lake size error estimation). The lake boundary/footprint size is evaluated by comparing the multi-spectral and backscatter threshold classifications within the inventory series itself, and in turn assessing the differences in classification. This produced an error estimate of  $\pm 0.77$  km2.

Finally, thank you for highlighting the resampling of the Landsat 8/9 Level 2 surface temperature data product. As a result, surface temperature estimates for each ice-marginal lake have been re-processed, using a 100 m pixel buffer to remove thermal contamination. The dataset, results and corresponding figures have been updated to reflect this. An interesting note is that the trends in surface temperature estimates remain unchanged, but generally lake surface temperatures were colder. In addition, we understand that results regarding the lake surface temperature estimates appear under reported compared to the other results sections. This was originally because we intended for these results to be reported and explored more thoroughly in a corresponding analysis publication. However,

it is appreciated that more details are needed here to balance the rest of the results section. To rectify this, an additional figure exploring spatial and temporal trends in lake temperature has been added (Figure 7), along with an expansion to Section 4.3 (Line 225-237). In all, this demonstrates a clear latitudinal trend in lake temperature as expected, and interesting links between lake size and temperature evolution.

**Minor comments**

Abstract – it may be beyond the scope of this style of paper but it could have more results in it? Some of the big lake decreases? Thermal results? Currently reads more like a short research proposal/rationale. Also would like to see '1km buffer' and 0.05 km2 in the abstract.

This was also a comment from Reviewer #1, so we have added more of the results/discussion to the abstract and removed some of the technical details.

20 – Greenland 'glacial' lakes were 21% of Zhang et al., (2024) global inventory...

The Zhang et al. (2024) global inventory only includes PGIC glacial lakes, not those associated with the Greenland Ice Sheet. Therefore, whilst this is a good piece of information, we will not include this in the manuscript.

20- St Pierre et al. (2019) argued proglacial lakes could be substantial CO2 sink – this inventory provides a big step to monitoring suspended sediment concentrations of them (contact and non-contact)

21 – as well as hydrological modification (higher temperature and lower SSC).

These are great additions that we are happy to add. The sentence is now updated: "The delayed release of meltwater at the ice margin is a significant, dynamic component of terrestrial storage, as well as a substantial CO2 sink and part of the hydrological system (St. Pierre et al., 2019)" (Line 24-25)

31 – Warren and Kirkbride (2003) needs to be cited here. Haresign and Warren (2005) and Roehl (2006) should be really as well.

Warren and Kirkbride (2003) and Röhl (2006) have been added to the references here (Line 35-36).

40 – and hydrological modification – key to know characteristics of water feeding into one of the most delicate parts of the thermohaline circulation...

Noted. Changes in hydrological conditions has now been added to this sentence: "This assumption overlooks the role of ice-marginal lakes as intermediary storage, and changes in lacustrine and hydrological conditions, caused for instance when glaciers retreat onto land." (Line 43-44).

53 – Can these be reclassified rather than retired? They're still important (see above).

See response to major comments.

**60 - And glacial lake response to climate?**

Yes, this has now been added to the sentence: "These inventories reveal evolving lake conditions that support future assessments of sea level contribution, lake response to climate change, ecosystem productivity, and biological activity associated with the Greenland Ice Sheet and the PGICs." (Line 61-63).

**Table 1 - Landsat 8/9 – filter – It should be 20% rather than 30%?**

Yes correct, thanks for catching this. It has now been changed to 20%.

Table 2 – Could you also add the following to improve useability further; i. centroid (could be from 2017) ii. Contact or non-contact iii. Temp\_time – time of Landsat image -> images in PM will likely capture daytime surface warming of very near surface layers (especially with high SSC)

Centroid positions have been added as the attribute "centroid". These are XY central coordinates for each lake, derived from all classifications across the inventory series. Coordinates are provided in the WGS NSIDC Sea Ice Polar Stereographic North (EPSG:3413) projected coordinate system.

Landsat image acquisition times have been added as the attribute "temp\_date", with acquisition datetimes formatted as "YYYY-MM-DD HH:MM". In the case that averages are composed of multiple Landsat images, each datetime is listed with a ", " break.

Contact and non-contact attributes have not been added, as outlined in the response to the major comments.

Adding dam type into the inventory would be desirable but not essential and clearly a huge amount of work that would be a whole different project in itself – probably requiring citizen science ground validation?

Dam type would be a valuable asset to the dataset, but like you say, would take a tremendous amount of work. In addition, dam type classifications may not be possible in scenarios such as consistent snow/ice cover, or in cases where high spatial resolution satellite imagery is needed to make a confident classification. Citizen science ground validation could aid in a small proportion of dam type classifications, as many ice-marginal lakes exist in unpopulated areas where this would not be possible (as reflected in the fact that most lakes do not have a corresponding placename).

103 – I would like to see '1km buffer' and '0.05km2' in the text here and also in the Abstract as they are key defining parameters of the inventory.

This information is defined in Section 5.1 (Data quality control); however, we understand the need for this to be defined earlier in the methodology section. The beginning of the methodology section has been updated to reflect this:

"Lake classifications (>=0.05 km2) were based on those adopted..." (Line 109)

In addition, this information has been added to the abstract (Line 6). We have chosen not to include the 1 km buffer as this is not a strict classification criterion. The 1 km ice margin buffer is an intermediary filtering step, after which lakes that do not share an ice margin are manually removed. This is described in Section 5.1 (Line 241-248).

128 – Nice strategy – this has been a key problem for a while especially with differing geology and consequently reflectance spectral signature from the lake SSC

Thanks very much.

**130 - 'best quality' - is this user defined? Or a class of product?**

The ArcticDEM mosaic version 3 was composed of the best quality strip data, which was manually defined. It appears this has changed for the version 4 release, where all strip data

is used in the compiling of the mosaic product. A reference to the version 3 mosaic product (Porter et al., 2018) has now been added to indicate where this information is sourced from: "Water bodies were classified from the ArcticDEM 2-metre mosaic (version 3), which is compiled from the best quality ArcticDEM strip files and manually adjusted to form a static data product (Porter et al., 2018)." (Line 138-139).

140 – 30 metre spatial resolution is incorrect – the original thermal pixels are 100m – so some pixels could easily be a combination of land (10 to 20 C) and lake water (4 C) with substantial thermal contamination. Either more needs explaining regarding the resampling method or stick with 100m to be on the safe side – there is plenty of data so can afford to lose some pixels.

As outlined in the response to the major comments, we have re-processed all lake temperature estimates with a buffer of 100 m and therefore this is reflected here, sticking to describing the 100 m resolution (Line 147) and modifying the border pixel description:

"Lake extents were cropped by a border pixel (i.e. 30 metres)..." >> "Lake extents were cropped by a border pixel (i.e. 100 metres)..." (Line 168-169).

142 – There needs to be more detail (Is it the NCEP reanalysis? 6 hour? 1 degree?) on the atmospheric data and correction used in this product given the nature of the study area and the Landsat L2 thermal product struggles in coastal zones (Dyba et al., 2022). I would imagine it handles the lower topography/stabler climate of SW Greenland better than the higher topography and dynamic climate and microclimates of E Greenland... (although I would still like to see how the regional average LSWT compared – see below)

According to the Landsat Atmospheric Auxiliary Data Format Control Book (2023), MODIS atmospheric auxiliary data and VIIRS atmospheric auxiliary data are used for Landsat 8/9 collection processing, with a geographic projection with 0.05 degree pixels (NCEP reanalysis is used for Landsat 4-7). Whilst Dyba et al. state that errors are larger in coastal zones, they also conclude that these can be corrected and the quality of temperature estimations for these zones can be improved. By validating and comparing in situ temperature data from Greenland (Section 5.4. Lake surface temperature error estimation), we present promising results that support this statement. In light of this, further information regarding the atmospheric auxilary data has been added to the corresponding methodology section, including a reference to the Landsat data format control book:

" along with ASTER datasets (global emissivity and normalised difference vegetation index) and MODIS and VIIRS atmospheric auxiliary data (geopotential height, specific humidity and air temperature) (Earth Resources Observation and Science (EROS) Center, 2020; Malakar et al., 2018; U.S. Geological Survey, 2023)." (Line 149-152).

**Did you eliminate temperatures below 0oC ? Needs adding in Methods.**

Yes, all estimates below freezing point (i.e < 0°C) were removed. This is now added to the methods section:

"Lake extents are cropped by a border pixel (i.e. 30 metres) to reduce the impact of edge effects." >> "Lakes extents were cropped by a border pixel (i.e. 100 metres) to reduce the impact of edge effects, and all unrealistic estimates below freezing (i.e. < 0 °C) were removed." (Line 168-169).

In addition, this information was added to the corresponding figure captions (Figure 6 and 7).

145 – Lakes in Poland are very different to glacial lakes in Greenland... (More stable atmosphere and less water input as well as lower SSC) So the comments on developing a calibration factor for Greenland in the Discussion are well founded. Using the GEE script from Ermida et al. 2020 would be more robust though? The validation data for SW Greenland does look very good though – which I think proves sufficient robustness.

Yes, lakes in Poland have differences to those in Greenland; however, this is the only study we found that convincingly derived water surface temperature estimations. They present several approaches to estimate water surface temperature, one of them being the application of a correction factor to the Landsat 8/9 reprocessed surface temperature science product. We compared in situ temperature measurements in Greenland to demonstrate a high confidence in applying this known correction factor. As stated though, we view this as the first step to defining a Greenland-specific lake correction factor – this endeavour will likely form a whole body of work in itself and is already yielding some interesting findings.

We had not come across the Ermida et al. publication and processing routines before, so thank you for making us aware of this. From reading the paper, it appears this is merely a land surface temperature estimation, rather than a water surface temperature estimation. However, we have cited the Ermida et al. work in the manuscript as we think it is valuable for readers to refer to: "...in degrees Celsius (NASA Applied Remote Sensing Training (ARSET) program, 2022; Dyba et al., 2022)." >> "...in degrees Celsius (Ermida et al., 2020; NASA Applied Remote Sensing Training (ARSET) program, 2022; Dyba et al., 2022)." (Line 159-160).

159 – I think the simplest way to deal with the thermal contamination at lake boundaries is to increase the buffer to 100m

Agreed. Lake temperatures have been reprocessed with a 100 m buffer now.

**163 – Add '... that have existed between 2016 and 2023'?**

Done. This was also highlighted by Reviewer #1. The sentence now reads as: "In total, the dataset identifies 2918 automatically delineated ice-marginal lakes across all inventory years (2016-2023) (Figure 1)." (Line 172).

173 – Yes I think the variability in abundance is impossible to study further at this scale – could be variations in meltwater flux, permeability of substrate (a large number may be permafrost underlain/ground ice – could potentially develop taliks...) dam porosity etc.

Agreed. We think that the change in the number of ice-marginal lakes needs to be studied on a decadal scale in order to identify meaningful trends. Given this is a living dataset that will continue to be added to year on year (and with a possibility of including prior years to the Sentinel era), it is hoped this will be possible in the future.

**192 – CE looks to be low variability in area in Fig. 3a? NO looks to be second largest variability in area on those results.**

Indeed, change in average lake area over the ice sheet margin is smallest at the CE ice sheet margin (0.30 km2). The largest change is experienced at the NO ice sheet margin (1.31 km2). Fluctuations in the average lake area at the PGIC margins are generally much smaller, apart from in the CE (2.20 km2) and NE (1.76 km2) regions. We have more clearly defined in the text when we are described Figure 3a (Ice Sheet lake change) and Figure 3b (PGIC lake change) (Line 177-186).

**195 – I think these declines in area are an important result. This should either be explored a bit further (maybe not appropriate in this paper) or flagged for future research etc (see comments above)**

Yes, as well as the many scenarios where lake size remains relatively consistent. We have added this to the section on future research:

"Tentative findings have been outlined, yet further analysis and evaluation against other datasets is needed to investigate causal links. For example, the inventory series could be used to address the drivers of change in lake area with comparison to potential influences such as meltwater flux, sedimentation rates, bedrock type, and GLOF magnitude and frequency (e.g. Veh et al., 2025)." (Line 299-302).

**201 – Yes I can see the glacier remnant on the NW side of the terminus. Some of the 'margin lines' on the South side look to be possibly from lake ice?**

We visually inspected the corresponding satellite imagery and it does appear that lake ice may be influencing the classified lake form in Figure 5a. In fact, this is an example of a lake with persistent ice cover throughout the summer (and for all inventory years). This has been noted in the figure caption:

"It is noted that the example from (a) is a lake with persistent ice cover throughout the summer season." (Figure 5 caption).

**203 – Important observations for 5c and 5d – worth flagging I think – are these a topic of ongoing research?**

Lacustrine terminus retreat dynamics at Greenland's ice margins is a topic that we think the ice-marginal lake inventory series could aid in investigating further. It has been investigated elsewhere, for example in Sweden (through your own work, Dye et al. 2021, 2022), in Patagonia (Minowa et al., 2017), and at the Cordillera Darwin (Langhamer et al., 2024). Lacustrine terminus dynamics have been examined in Greenland, but largely limited to the Kangerlussuaq (SW Greenland) region (e.g. Mallalieu et al., 2021). Consequently, the inventory series could support the study of lacustrine terminus retreat dynamics more widely across Greenland, perhaps on a regional scale, or more ambitiously on a national scale. We, ourselves, are not currently looking into this as a follow-on from the Greenland inventory series; however, we have added this to Section 6.1 (Uses for the ice-marginal lake inventory series): "Additionally, the inventory series would be a valuable dataset for examining lacustrine terminus retreat dynamics, expanding investigations from a case study basis (e.g. Mallalieu et al., 2021; Langhamer et al., 2024) to a regional and/or national scale (e.g. Dye et al., 2022)." (Line 304-306).

**Section 4.3 – This section currently is under explored/reported.**

As noted in the response to major comments, we have recitfied this by including a figure (Figure 7) exploring spatial and temporal trends in lake temperature has been added, along with an accompanying expansion of Section 4.3 (Lake surface temperature) (Line 225-237). In all, this demonstrates a clear latitudinal trend in lake temperature, and interesting links between lake size and temperature evolution.

209 – Is this the average for all pixels? Or Sum of all lake averages divided by number of lakes? (some of the large lakes could skew this)

**211 – of all pixels?**

These values represent the sum of all lake averages divided by the number of lakes. This has been clarified in the text (Line 222-223) and caption for Figure 6:

"Examining the average lake surface temperature estimate across all lakes, the average lake..." >> "Examining the average lake surface temperature estimate across all lakes (i.e. the sum of all lake averages divided by the number of lakes), the average lake..." (Line 222-223).

As said above, this is the sum of all lake averages divided by the number of lakes. This is added to the text and corresponding figure caption.

212 – Replace 'falling' with 'being lower' as these are snapshots

Done.

214 – Replace 'rising' with 'being higher'

Done.

There is more room for exploration of the thermal results – either here or signposted to a future publication. If the thermal contamination issues are resolved, the average lake temperature by region should be shown. Also potential for average temperature by different lake size classes etc. With higher confidence in the data the lower temperatures in 2018 could be explored further too. At the moment the thermal data feels a bit like a 'bolt on' component.

See response to major comments (also detailed in the feedback for Section 4.3) as to how the thermal results have been expanded upon with, for instance, lake size compared to average surface temperature.

224 – Again can these detach lakes be reclassed as non-contact or put in a separate archive? See response to major comments.

**247 – Boundary error/uncertainty estimations of some kind need including here.**

Agreed. This was also noted by Reviewer #1. In response, we have added a section on the lake size error estimation (Section 5.3). Lake boundary/footprint size is evaluated by comparing the multi-spectral and backscatter threshold classifications, producing an error estimate of  $\pm 0.77$  km2.

258 – Agreed estimates are reliable. The validation data has an interesting cluster of points around the lake sensor temperature of 4 oC – could be afternoon warming of surface water? Time of day of image capture is important.

Yes, this is an interesting cluster that we would like to explore further (possibly as a designated project on lake temperatures in Greenland). This cluster originates from two specific lakes - Qamanersuaq and ST924. In both these instances, all measurements and coinciding remote sensing estimates are from afternoon acquisitions (between 13.00-15.00). However, this is the case for all of the validation dataset presented here – all measurements coincide with afternoon Landsat satellite passes. Therefore, it is likely that this clustering is associated with another influencing factor, such as lake depth/morphology or suspended sediment concentration. We have added this information to figure caption (Figure 9) and the manuscript to clarify this:

"All observations are fro afternoon acquisitions (between 13.00-15.00 UTC)." (Figure 9 caption).

"An interesting cluster of data points is evident, originating from measurements taken at Qamanersuaq and Asiaq station 924 which could be related to specific lake characteristics, such as lake depth/morphology or suspended sediment concentration." (Line 289-291).

265 – Yes this is a really important inventory for assessing how glacial lakes form part of a deglaciating land system and how their response to climate affects hydrology and ecology downstream. At the moment this section reads too glaciology focused – there is much wider scope for assessing the glacial lake evolution. How will lake development affect downstream sediment budgets? How will temperature changes affect ecology? (Fellman et al., 2014)

Remarks on wider applications of the dataset have been added to Section 6.1 now. See comment from Line 203 for full details.

**292 – Are there any TanDEMX products that would help? (Is access through ESA possible?)**

TanDEM-X could be an option, however, Lutz et al. (2024) stated that coverage over Greenland is sporadic due to the acquisitions being campaign-based. We would need to ensure that coverage could at least yield an annual Greenland-wide DEM for sink detection classifications, however, we have not looked further into the feasibility of this. TanDEM-X has been added alongside the ArcticDEM strip data as a potential DEM dataset to use in the future:

"For the DEM classification, an alternative would be SAR-derived DEMs from the TanDEM-X mission or the ArcticDEM strip data product, which are both time variant, but data coverage is lacking currently and scenes covering all Greenland may not be possible from year to year (e.g. Lutz et al., 2024)." (Line 332-334).

**Figure 1 – More thermal results need to be added**

Average surface temperature for each region is now provided in the regional table statistics, replacing the largest lake size (as this is provided in the annotations on the map identifying the largest lakes for each region).

**Figure 3 – NE lake area was high for 2018**

Yes, it is markedly high. This has now been noted in the corresponding text (in Section 4.2):

"Average lake size is highest in the NE region in 2018 and 2022, with an average size of 2.71 km2 and 2.77 km2, respectively..." Line 200-201).

*Figure 4 – Currently difficult to distinguish between the ocean and ice sheet. I suggest having ocean in a shade of blue.*

Done.

Figure 5a – See above. I would like to query the lake margin for 2017 in Figure 5a – which looks to have a pattern suggesting lake ice at the margins?

Done, the ocean area in the overview map has been changed to a shade of blue. We visually inspected the imagery from 2017 and it does appear that lake ice may be influencing the classified lake form in this case. In fact, this is an example of a lake with persistent ice cover throughout the summer (and for all inventory years). This is now noted in the figure caption:

"It is noted that the example from (a) is a lake with persistent ice cover throughout the summer season." (Figure 5 caption).

Figure 6 – data filtered below 0C? (If so please add this in the caption) Looks like lots of noise > 8 C

Yes, all estimates below 0°C are filtered out. This has now been added to the figure caption:

"Averages are calculated from all available scenes acquired from the month of August to limit the risk of mis-estimates due to ice-covered conditions, with all estimates below 0°C removed." (Figure 6 caption).

---

## Referee Report (RR1)

**Review of "The Greenland Ice-Marginal Lake Inventory Series from 2016 to 2023" (essd-2025-18) by Penelope How and others**

**Summary**

The authors present a time-varying inventory of ice-marginal lakes across Greenland that extends a past static 2017 inventory. The authors rely on this earlier work heavily for the present methods, but the methods are generally well described. However, the authors treatment of icebergs in the lake was unclear in the current version, which could have significant implications for the interpretation of lake surface temperature and area results described here. While I described several "major" issues below and numerous minor points, I consider my comments to reflect a suggestion of "minor revisions", as everything can be addressed by text better describing methods, clearly acknowledging potential issues/data artifacts, and a a few new analyses of the existing inventory (rather than anything requiring revision of the inventory itself).

**Major comments**

Sec 3.2: from this section and inspection of Figure 5, it doesn't seem like you are masking out icebergs from your water temperature estimates, is that right? If I'm wrong, please try to more clearly state this. If you aren't masking icebergs, how much of an effect could this have on your temperature estimates?

L243: Do you have any idea of the prevalence of false positives vs false negatives? If I understand this sentence correctly, you are taking data from a false positive rate (i.e., automated method says there's a lake, but there's not) and are then using this value as a uniform  $\pm$  error. But do you know if the method misses lakes just as often as it makes them up? It seems like this could lead to substantially different errors on the positive vs. negative sides.

Also, it would be good to have an estimate of error in lake area and its change. I imagine the error in lake area is much smaller (in % terms) than the error in number that you discuss here. Presenting both of these errors could allow you to say, "while absolute number is somewhat uncertain, uncertainty in lake area change is small (if true), suggesting that the lake number error is primarily attributable to varying detection of small lakes". That would help bolster the utility of this dataset, which I imagine could be undermined by only reporting the  $\pm 36\%$  error in lake number.

Figure 3: For bottom plots > the average lake area is far noisier than I'd expect given the many lakes being used to compute these averages. What do you attribute these large changes to? To me, the first thing that comes to mind is data processing artifacts like large lakes being split into multiple pieces in some years. Without a commentary on what underlies the high variability, it is hard to know how much to trust it as a physically meaningful value vs. data noise.

Figure 5: From looking at this image, it doesn't seem like you're doing any hole filling to remove icebergs floating in the lake? This seems like it could impose substantial variability in a lake's area following large calving events > can you comment on this somewhere?

Figure 6: Given the high noise level here, some measure of uncertainty/variation (perhaps interquartile range) would be helpful for ascertaining whether the plotted changed in the mean are due to real variation vs. noise. To me, this data seems like it would be better shown in box plot (or violin plot) form, so we could get some sense of the distribution of data, which is not easily grasped at present (aside from seeing that there is a lot of variation from lake to lake).

**Minor comments**

L39: The Shugar paper doing global ice-marginal lake mapping made an estimate for meltwater retention in lakes > it should probably be mentioned even if there are reasons why the estimate is imperfect (that can also be mentioned)

L73: Is this the GLO90 DEM, or what static DEM are you referring to?

Sec 2.2: I am not sure if this section is required by ESSD, but I personally didn't get much out of it (think most will be repeated later in more detail?) and think the whole thing could be deleted. Otherwise, the utility/value of this section should be made clearer.

Table 1: I suspect you mean 20 and 30% cloud cover limit across the whole image, not on some kind of pixel-wise basis > is that correct? It could be worth spelling this out

L129: It is not clear to me what you mean by "where positive classifications adhere to all thresholds" > please reword or clarify

Table 3: It would be nice to include the names of each band (e.g., "B2 = blue") in the caption for people less used to working with Sentinel data. I think maybe you are saying that it is considered water if it's less than that threshold value?

L168-178: Do you have a sense how much of this variation in lake number is due to processing artifacts (e.g., one lake classified as two in some years due to data issues) as opposed to physically meaningful variations? It seems like you could do some kind of intersection/spatial join to test this.

L195: It is interesting that many more lakes shrunk in size > this seems at adds with what is seen in many areas. Are you doing this analysis on a last year- first year basis? This would make your results really sensitive to noise in those years. Have you tried doing a linear fit to all lake area data at a site and making this threshold based off a rate of change rather than an absolute change? That

seems like it would be more resistant to noise. Regardless, do you have any physical interpretation of why shrinking lakes are more common? Maybe this comes later.

Sec 4.3 (related to Figure 6 comment): please discuss if icebergs are masked during temperature estimates. If not, it is unclear what these data mean. Also, have you analyzed lake temperature trends (or year-to-year temperature variability) on a lake-by-lake basis? This seems like it would add a lot to this section. As is, it is a little unclear what the lake surface temperature adds to the story here.

L229: I imagine this is described in more detail in How 2021, but it is unclear in the current manuscript how the different delineation methods are incorporated. How do you blend the datasets when they have inevitably somewhat differing shapes? In general, how do you choose whether just one or multiple methods are used to delineate a given lake?

Sec 6.1: I think this is all true, but much applies to a time static lake inventory, so it might be useful to better articulate what having the time variation adds here.

---

## Author Response (AR2)

**Response to Editor**

I am pleased to see that the authors have been able to incorporate most of the reviewers' recommendations. In addition to those comments, I would like to raise a major concern regarding the error assessment. Specifically, assuming a fixed error of 0.77 km² for all lakes—which appears to be half the absolute difference between two mapping approaches—seems somewhat arbitrary. For instance, for half of all lakes (

Median statistics are also provided in the corresponding results section (Section 4.2: Lake surface extent). And statistics presented in Figure 3 refer to change in median lake area across the inventory years.

L25: Please mention here what an ice-marginal lake is according to your definition (I notice that it is mentioned later in the manuscript, but it would be clear about this definition very early in the manuscript)

Done. The definition of an ice-marginal lake is elaborated on, and introduced earlier in this section.

"...along the ice edge of the Greenland Ice Sheet and in front of, and beside, surrounding PGICs. The delayed release of..." >> "...along the ice edge of the Greenland Ice Sheet and in front of, and beside, surrounding PGICs. An ice-marginal lake is a reservoir of meltwater that is dammed by a moraine or by the ice itself, therefore the trapped meltwater is in contact with the ice margin. The delayed release of..." (Line 24-25).

L32: There is probably little evidence for (recent) megafloods in Greenland, so I suggest to remove this hint.

This passage has now been removed.

L72: This buffer is a bit confusing: why is there a 1-km buffer, if lakes need to be in direct contact to glaciers according to your definition above?

One of the limitations of our dataset production is the lack of a dynamic, time-resolved ice margin dataset. We rely on the GIMP ice margin, which was produced in 2019 and does not reflect changes over time. If we applied this ice margin without a buffer, we would effectively be assuming a static ice margin across all inventory years, which is not realistic. The 1-km buffer is therefore used as an initial spatial filter to account for glacier retreat and to remove lakes that are very unlikely to have been in contact with the ice margin at any point. After this automated step, we manually inspect and remove lakes that are clearly detached from the ice margin. This is now detailed in Line 162-164, Section 3.1.4 (Inventory compilation).

L89: Please clarify in the manuscript that several lake polygons within the same year can share the same ID. In addition, specify the criteria used for assigning new IDs: Is there a minimum distance threshold between polygons that triggers the assignment of a new ID, or is a new ID assigned whenever two polygons do not spatially overlap?

Multiple lake polygons within the same year can share the same ID. This occurs when polygons have been connected at some point in time in the inventory series, even if they appear disconnected in a specific inventory year. Assigning the same ID in such cases preserves temporal continuity and reflects the dynamic nature of lake evolution, including drainage and partial refilling events. ID assignment is performed manually based on visual inspection across years; therefore, there is no fixed minimum distance threshold or

automated rule. Automated approaches were tested but did not reliably capture the complexity of lake connectivity over time. We have clarified this in the manuscript (Line 174, Section 3.1.4: Inventory compilation).

L93: In checking the GIMP GeoTiffs, I wondered how this dataset allows you distinguishing whether a lake is touching an ice cap, ice sheet or peripheral glacier?

The GIMP GeoTiffs are merged and transformed into a vectorised (.shp) format, from which we can distinguish lakes that are in contact with the GrIS and/or a PGIC. This information has now been added to the manuscript, along with the acronym definitions for MEaSUREs and GIMP:

"This margin information originates from the MEaSUREs (NASA Making Earth System Data Records for Use in Research Environments) GIMP (Greenland Ice Mapping Project) 15 m ice mask, where the provided GeoTiff files were merged and transformed into a vector format (Howat et al., 2014; Howat, 2017)."

L106: This point shapefile should ideally indicate which of the lakes were not detected by the algorithm.

This is a good idea and has now been added to the dataset point geopackage (.gpkg) file. The new version is now released on the GEUS Dataverse (https://doi.org/10.22008/FK2/MBKW9N), with files denoting a new version (fv3).

3.1.1 – 3.1.3: We understand that your multi-temporal analysis is based on the algorithm proposed by How et al. (2021), and that you refer extensively to the methods outlined in that study. However, for a data publication in ESSD, traceability and reproducibility are essential. We therefore kindly ask you to expand the methodological sections in your manuscript to make the workflow fully understandable without requiring the reader to consult external sources.

We have added methodology information where possible to the subsections of Section 3.1 (Lake classification) in order to improve the traceability and reproducibility of the classification methods. In addition, we have added a subsection detailing how the inventory was compiled (Section 3.1.4: Inventory compilation), which includes filtering approaches and the subsequent manual curation. Metadata generation is referred to here, but the majority of the metadata production information remains in Section 2.3 (Data format and structure).

For clarification, the SAR backscatter classification approach is simplified because of its migration to Google Earth Engine and therefore does not require as much detail (compared to How et al., 2021). The multi-spectral indices classification approach is summarised in Table 3 (Summary of multi-spectral indices for ice-marginal lake classifications from Sentinel-2 Level 1C scenes), including thresholds and indices targets. As a result of these aspects, we felt that the subsections regarding the SAR backscatter and multi-spectral indices classification approaches were adequately refined whilst also retaining all necessary information for the reader to understand the methodology.

L163: Where can we see that the correction factor appears to agree well with the limited datasets available?

This passage is eluding to results presented in the manuscript, therefore we have moved it to the appropriate place in Section 5.4 (Lake surface temperature error estimation) at Lines 307-210.

L190: Romer Soe needs a coordinate.

Coordinates (in degrees, minutes and seconds; DMS) have now been added:

"The largest lake in the inventory is Romer Sø, located in northeast Greenland..." >> "The largest lake in the inventory is Romer Sø ( $80^{\circ}59'54"N$ ,  $19^{\circ}09'21"W$ ), located in northeast Greenland..." (Line 190).

L190-195: Most studies focusing on lake area change report the total (summed) lake area within a given region, including the errors in total lake area, and we would appreciate if you could provide similar summary statistics. Given the potential influence of outliers (very large lakes such as Romer Sø), we suggest reporting the median lake area as a more robust diagnostic for comparing different study regions.

Total (summed) lake area and median lake area now included in Figure 3, where the total ice sheet lake area and total PGIC lake area are provided alongside lake abundance and median lake area. In addition, the associated text has been updated in Section 4.2 (Lake surface extent) describing changes in total lake area and median lake area across regions and through time (Line 224-237).

Additionally, we would like more clarity on how the different methods contributed to the estimated lake surface area. If we understand correctly, you obtain between one and three separate estimates of lake area per lake. How are these individual estimates combined to form the reported mean? For instance, are the estimates averaged per lake, or are the lake areas first dissolved across methods before statistics are derived?

As you describe, lake areas are first dissolved across methods and then the statistics are derived for comparing changes in lake area through the inventory years. This is now outlined in Section 4.2:

"The inventory series also holds information on the change in lake area over time, by comparing corresponding lake extents from one of the direct classification methods (i.e. from SAR and/or multi-spectral imagery) (Figure 3). Lake areas are first dissolved across the SAR and multi-spectral classifications, and then statistics are derived by comparing lake area through the inventory years. Change in average lake area over..." (Line 217-220)

L253: Could you please clarify in more detail how you ensure that the algorithm performs robustly across the entire time series, given that the static DEM—which represents only a single point in time—appears to contribute significantly to the derived lake areas? Specifically, how do you account for potential changes in topography over time, e.g. in situations when glacial retreat creates a larger basin, that may affect the accuracy of lake area delineation when relying on a fixed elevation reference?

We outline that the ArcticDEM does not directly detect water (Section 3.1.1, Line 147-149) and therefore lakes classified using the DEM sink classification approach are not used in the lake area analysis presented in this manuscript (as stated in Section 4.2, Line 226-232). Manual verification of each detected lake is performed for each inventory year in order to remove any topographic depressions that are not filled and to remove lakes that are detached from the ice margin (as stated in Section 3.1.4, Line 167-176).

L254-256: Please revise this sentence for clarity.

We have changed this sentence accordingly to better convey this:

"This is likely because the classification methods have been extensively applied and developed in the SW region compared to others..." >> "This is likely because the classification methods have been extensively applied and developed in the SW region, making them particularly well-suited for use there compared to other regions" (Line 270-271)

5.2./ 5.3 We would appreciate it if you could focus this paragraph more explicitly on the error estimates presented in the current study, rather discussing the method/ results from the earlier study. As stated above, we encourage you to provide error estimates at the level of individual lake areas, rather than reporting only a single aggregated or gross estimate across all lakes.

For Section 5.2, we have removed information regarding the error estimation from How et al. (2021).

For Section 5.3, the error estimation for lake area has been updated to better quantify error based on total lake area (rather than an average individual lake area error). Our analysis shows that the total absolute error (i.e. the total difference between SAR lake area and multi-spectral lake area classifications) is  $774.94 \, \mathrm{km^2}$ , equating to a central percentage error of  $\pm$  5%. We have included a summary of these error statistics (Table 4) showing this reported total lake area error, along with lake area error based on lake size groupings. We included this to show that, whilst the absolute error propagates with the size of a lake, this does not affect the relative (%) error. The text in this section has been updated accordingly (Lines 289-301).

Additionally, we are unclear about your error estimate for lake abundance. If we understand correctly, your results suggest a negative bias—that is, an underestimation of lake numbers—yet your reported error range includes both negative and positive values. Could you please clarify the rationale behind this, and explain whether this range represents uncertainty around a central estimate, or some other form of error characterization?

We form the lake abundance error estimation around a central estimate. We have clarified this in the text and included a passage to account for why the dataset under-estimates icemarginal lakes in Greenland:

"Across all inventory years, 4543 ice-marginal lakes were manually identified in total, of which 2915 (64%) are captured by the automated classification approach. This forms a central error estimation of  $\pm$  809 (36%), and demonstrates that the automated classifications in the ice-marginal lake inventory series underestimate the number of ice-marginal lakes across Greenland. However, manually classified lakes include those under the size threshold (i.e. <0.05 km²) adopted in the automated classification approach. The under-estimation of ice-marginal lakes within the inventory series therefore, in part, reflects smaller lakes that are removed from the dataset automatically due to the minimum area filtering. To summarise, lake abundance in the ice-marginal lake inventory series

| should be adopted as a conservative estimate as it does not account for lakes with a surface extent below $0.05~\rm km^2$ ." (Line 280-285) |  |
|---------------------------------------------------------------------------------------------------------------------------------------------|--|
|                                                                                                                                             |  |
|                                                                                                                                             |  |
|                                                                                                                                             |  |
|                                                                                                                                             |  |
|                                                                                                                                             |  |
|                                                                                                                                             |  |
|                                                                                                                                             |  |
|                                                                                                                                             |  |
|                                                                                                                                             |  |
|                                                                                                                                             |  |
|                                                                                                                                             |  |
|                                                                                                                                             |  |
|                                                                                                                                             |  |
|                                                                                                                                             |  |
|                                                                                                                                             |  |
|                                                                                                                                             |  |
|                                                                                                                                             |  |
|                                                                                                                                             |  |
|                                                                                                                                             |  |
|                                                                                                                                             |  |
|                                                                                                                                             |  |
|                                                                                                                                             |  |

---

## Author Response (AR3)

**Response to Reviewer #1**

The authors present a time-varying inventory of ice-marginal lakes across Greenland that extends a past static 2017 inventory. The authors rely on this earlier work heavily for the present methods, but the methods are generally well described. However, the authors treatment of icebergs in the lake was unclear in the current version, which could have significant implications for the interpretation of lake surface temperature and area results described here. While I described several "major" issues below and numerous minor points, I consider my comments to reflect a suggestion of "minor revisions", as everything can be addressed by text better describing methods, clearly acknowledging potential issues/data artifacts, and a few new analyses of the existing inventory (rather than anything requiring revision of the inventory itself).

Thank you for your feedback and comments on the revised manuscript. We have addressed these minor revisions below, of which we highlight the main updates here:

- 1. The average lake surface temperature is now presented with box plot analysis in order to provide a better evaluation on data trends and data variability
- 2. The rate of lake area change is updated with linear regression slope calculations, which provides valuable insights into the relationship between the rate of change and lake size
- 3. Clarification on iceberg removal from the average lake surface temperature estimates has been provided, which is performed implicitly through strict scene acquisition selection and the removal of temperature estimates that do not represent open-water conditions (i.e. values below  $0^{\circ}$ C)
- 4. Median and total (summed) lake area statistics have been incorporated into the manuscript to improve the overview of the inventory series, and provide more insight into the regional- and annual- comparisons
- 5. Additional technical details have been provided to clarify how lake delineations from multiple classification methods are merged. In addition, an example tutorial of this merging (in the form of a Jupyter Notebook tutorial) is now provided with the source code used to produce the inventory series

Please find more information regarding these changes below, along with individual responses to the line-by-line feedback.

**Major comments**

Sec 3.2: from this section and inspection of Figure 5, it doesn't seem like you are masking out icebergs from your water temperature estimates, is that right? If I'm wrong, please try to more clearly state this. If you aren't masking icebergs, how much of an effect could this have on your temperature estimates?

The dataset represents an annually averaged inventory series designed to ensure high confidence and full spatial coverage across Greenland. Water temperature estimates are derived from the

detected lake extents for each inventory year. These extents are based on averaged scene acquisitions from July and August to minimize the likelihood of iceberg and ice cover presence.

The classification of icebergs and ice cover across lakes in the Arctic remains challenging (e.g. Carrea et al., 2025; Dye et al., 2025), and no robust, transferable classification currently exists for the entire region. To address this, we adopt an alternative approach by removing temperature estimates below 0  $^{\circ}$ C, as these do not represent open-water conditions. This effectively limits the influence of icebergs and ice cover on the derived water temperature estimates.

We acknowledge that explicit masking of icebergs and ice cover would further improve temperature accuracy. We therefore identify this as an area for future work, starting with detailed analyses of lake surface temperature evolution at selected sites or regions. However, implementing such masking at a Greenland-wide scale is currently beyond the scope of this dataset.

To clarify this, we have elaborated on the approach to reducing the influence of icebergs and ice cover in the corresponding methodology section (Section 3.2 Summer water surface temperature estimation):

"A summer average water surface temperature estimate was derived using this approach, calculating an average, maximum and minimum water surface temperature value for each lake extent over each inventory year, along with the standard deviation. Scenes were filtered by a maximum cloud cover of 20%, with acquisitions limited to the month of August to reduce the probability of ice-covered lake conditions. Lake extents were cropped by a border pixel (i.e. 100 metres) to reduce the impact of edge effects. All unrealistic estimates below freezing (i.e. 0 °C) were removed, therefore reducing the influence of icebergs and ice cover on the water temperature estimates." (Line 193-198)

In addition, we have added information to the corresponding error estimate section (Section 5.4 Lake surface temperature error estimation) to outline the scope of the methodology and discuss the handling of icebergs and ice cover:

"In addition, the influence of ice presence is limited based on the selection of strictly summer scene acquisitions and removing temperatures below 0 °C. In future work, icebergs and ice cover could be explicitly removed before a temperature estimate is derived to reduce the influence of their presence. This is currently beyond the scope of the dataset given the challenges with classifying the presence of ice on lakes across large regions in the Arctic (e.g., Carrea et al., 2025; Dye et al., 2025)." (Line 339-346)

L243: Do you have any idea of the prevalence of false positives vs false negatives? If I understand this sentence correctly, you are taking data from a false positive rate (i.e., automated method says there's a lake, but there's

not) and are then using this value as a uniform  $\pm$  error. But do you know if the method misses lakes just as often as it makes them up? It seems like this could lead to substantially different errors on the positive vs. negative sides.

In our analysis, all lakes were manually identified and mapped as point locations, with these manually verified data points forming the foundation for the error estimate. The automated classification method is designed to be robust, with no false positives (lakes incorrectly identified by the automated method). This is ensured by a thorough manual verification and intervention process that removes any misclassified lakes from the dataset. Therefore, the automated method may only exclude lakes from the inventory (i.e., false negatives), but it does not introduce any false positives.

The error estimate of  $\pm 809$  lakes (36%) in the manuscript represents a conservative lower bound. This underestimation primarily reflects false negatives, where lakes are missed by the automated method. It is also important to note that the undercounting is partly due to the size threshold (lakes smaller than  $0.05 \text{ km}^2$ ) used in the automated classification, which excludes smaller lakes from the dataset.

Given this, we agree that the error estimate should be described as a conservative lower bound rather than a uniform error estimate. We have updated the text to reflect this:

"Across all inventory years, 4543 ice-marginal lakes were manually identified in total, of which 2915 (64%) are captured by the automated classification approach. This forms an abundance error estimation of  $\pm$  809 (36%), which reflects the undercounting of lakes by the automated classification. However, manually classified lakes include those under the size threshold (i.e. <  $0.05~\rm km^2$ ) adopted in the automated classification approach. The under-estimation of ice-marginal lakes within the inventory series therefore, in part, reflects smaller lakes that are removed from the dataset automatically due to the minimum area filtering. Therefore, the error estimate reported is a conservative lower bound, which reflects the underestimation of ice-marginal lakes due to false negatives (i.e. where lakes are missed by the automated classification method)." (Line 322-328)

Also, it would be good to have an estimate of error in lake area and its change. I imagine the error in lake area is much smaller (in % terms) than the error in number that you discuss here. Presenting both of these errors could allow you to say, "while absolute number is somewhat uncertain, uncertainty in lake area change is small (if true), suggesting that the lake number error is primarily attributable to varying detection of small lakes". That would help bolster the utility of this dataset, which I imagine could be undermined by only reporting the ±36% error in lake number.

An estimate of lake area is included in the manuscript that addresses this comment. Specifically, extents classified with the SAR and multi-spectral methods provide a proxy for estimating error by comparing extent variability. On average, there was a difference of 1.54 km2 (median: 0.17 km2)

between SAR and multi-spectral delineations. When all lake areas were summed and compared, there was a total lake area error of 774.94 km². This error propagates according to lake size, with lakes over 5.00 km² having a total error of 405.98 km² while lakes under 0.10 km² had a total error of 1.89 km². The reported abundancy error estimate (± 809, 36%) is substantially larger. Therefore, as you say, "while absolute number is somewhat uncertain, uncertainty in lake area change is small, suggesting that the lake number error is primarily attributable to varying detection of small lakes". This was added to the manuscript based on feedback from a previous reviewer, which is presented in Section 5.3 (Lake size error estimation). The reported error estimate for lake extent/area includes a table summarising the error propagation with lake size (Table 4).

Figure 3: For bottom plots > the average lake area is far noisier than I'd expect given the many lakes being used to compute these averages. What do you attribute these large changes to? To me, the first thing that comes to mind is data processing artifacts like large lakes being split into multiple pieces in some years. Without a commentary on what underlies the high variability, it is hard to know how much to trust it as a physically meaningful value vs. data noise.

In connection with previous reviewer feedback, we attribute the "noise" in Figure 3 to using mean (average) statistics, which are largely influenced by outliers (e.g. large lakes such as Romer Sø). We now present median statistics to address this concern, and provide a more robust diagnostic for our results and analysis. Total (summed) lake area and median lake area now included in Figure 3, where the total ice sheet lake area and total PGIC lake area are provided alongside lake abundance and median lake area. In addition, the associated text has been updated in Section 4.2 (Lake surface extent) describing changes in total lake area and median lake area across regions and through time (Line 222-228).

With regards to processing artefacts, we demonstrate that such artefacts are largely removed from the dataset in response to concerns on variations in lake abundance (L168-178). Specifically, an intersection routine is performed to remove processing artefacts, where lakes are grouped consistently across all inventory years. This is to combat cases where a lake separates into two water bodies – in our production pipeline, such instances will still be classified as one lake across the inventory series.

Figure 5: From looking at this image, it doesn't seem like you're doing any hole filling to remove icebergs floating in the lake? This seems like it could impose substantial variability in a lake's area following large calving events > can you comment on this somewhere?

This is correct. We do not perform hole filling, as these areas may represent both consistent features (e.g. islands) and transient features (e.g. icebergs). Applying an automated hole-filling routine could therefore introduce additional errors by artificially modifying lake extents. Instead,

we have chosen to retain the natural variability observed in the classified lake boundaries, acknowledging that iceberg presence may contribute to short-term variability in lake area, particularly following major calving events. We now clarify this in the corresponding results section (Section 4.2: Lake surface extent):

"The lack of a strong trend could reflect the high climatic variability across Greenland given its large latitudinal range. It could also be related to variability in lake contact with the ice margin over time, and/or varying iceberg presence which is not considered in the SAR and multi-spectral classification approaches." (Line 237-239)

The caption in Figure 5 has also been updated to described what holes in the polygons shown can represent:

"Figure 5. Examples of lake morphology change, and the corresponding evolution of ice termini morphology, from the ice-marginal lake inventory series. These examples highlight basin margin retreat (a), peripheral margin retreat (b), bilateral margin retreat (c), and focused margin retreat (d). It is noted that the example from (a) is a lake with persistent ice cover throughout the summer season. Holes in the lake delineations are attributed to consistent features, such as islands, and transient features, such as icebergs. The background satellite imagery presented is from a Sentinel-2 10 m 2022 mosaic (Styrelsen for Dataforsyning og Infrastruktur, 2024). The base layers for the insert map plotting are from QGreenland v3.0 (Moon et al., 2023)"

Figure 6: Given the high noise level here, some measure of uncertainty/variation (perhaps interquartile range) would be helpful for ascertaining whether the plotted changed in the mean are due to real variation vs. noise. To me, this data seems like it would be better shown in box plot (or violin plot) form, so we could get some sense of the distribution of data, which is not easily grasped at present (aside from seeing that there is a lot of variation from lake to lake).

We have updated Figure 6 to include box plots that show the median and interquartile ranges (25th–75th percentiles) of lake surface temperatures for each inventory year. This revision helps better represent the distribution of the data and the variability across the different lakes. The addition of these box plots provides a clearer picture of the data variation and allows for more precise evaluation of the trends.

Upon examining the box plots, we observe that the interquartile ranges and the overall distribution of lake temperatures remain relatively stable across the inventory years. This suggests that there have been no major shifts in data collection methods or sample size that could have introduced substantial variability. While the figure shows some fluctuation in lake surface temperatures, no clear long-term trend in the average temperature is evident. However, there is a notable anomaly in 2018, where the average lake surface temperature drops to its lowest value (3.8°C). This anomaly could reflect a real environmental influence or climatic event; however, we

acknowledge that it is also possible that this could be a data processing artefact. Further in-depth analysis would be required to determine whether the anomaly is genuine or an artefact. However, as this manuscript focuses on dataset description, this kind of detailed analysis is beyond the scope of the current study.

We have highlighted this when discussing the Figure in Section 4.3 (Lake surface temperature):

"The median lake surface temperature follows a similar trend, fluctuating between 3.0 °C (2018) and 4.4 °C (2023). Fluctuations year on year vary, with instances of lake temperature being lower between annual time steps (e.g. from 4.5 °C to 3.8 °C from 2017 to 2018), higher (e.g. from 4.8 °C to 5.3 °C from 2022 to 2023), and remaining consistent (e.g. 4.8 °C for 2021 to 2022). Overall, there is no evident trend in average lake surface temperature across the period. However, there is a notable anomaly in 2018 which could reflect a true climatic event but needs to be investigated further."

Additionally, the caption to Figure 6 has been updated to describe the box plots:

"Figure 6. Average lake surface temperature estimates from the month of August at each inventory lake for each inventory year (2016-2023) (grey). The average of all lakes (black) is the sum of all lake averages divided by the number of lakes, corresponding to the values reported on the plot. Boxplots (blue) indicate the median and the interquartile range (25-75%), with the shaded band representing the variation across the annual interquartile ranges..."

**Minor comments**

L39: The Shugar paper doing global ice-marginal lake mapping made an estimate for meltwater retention in lakes > it should probably be mentioned even if there are reasons why the estimate is imperfect (that can also be mentioned)

Done. We have highlighted the potential impact on terrestrial meltwater retention, in reference to Shugar et al. (2020):

"With continued retreat of the Greenland Ice Sheet under a warming climate, ice-marginal lakes are expected to become more abundant, larger and warmer; and will likely amplify lacustrine-driven proglacial melt rates, GLOF events, and terrestrial meltwater retention (Carrivick and Tweed, 2016; Grinsted et al., 2017; Shugar et al., 2020; Carrivick et al., 2022; Dye et al., 2021; Dømgaard et al., 2023; Lützow et al., 2023; Rick et al., 2023; Veh et al., 2023; Holt et al., 2024; Zhang et al., 2024)." (Line 38-42)

L73: Is this the GLO90 DEM, or what static DEM are you referring to?

The static DEM is in reference to the ArcticDEM 2-metre mosaic, which is used in this study to perform the topographic sink classification of ice-marginal lakes. As suggested in the next comment, Section 2.2 has now been moved and integrated with the relevant methodology sections. As a result, the DEM dataset is first referred to in Section 3.1.3 (Sink classification) now, removing this ambiguous reference.

Sec 2.2: I am not sure if this section is required by ESSD, but I personally didn't get much out of it (think most will be repeated later in more detail?) and think the whole thing could be deleted. Otherwise, the utility/value of this section should be made clearer.

Section 2.2 (Data sources and acquisition) is not specified in ESSD's requirements, but data sources/acquisition have to be clearly outlined, hence why we chose to include the section. We understand that there is repetition here with the subsequent methodology sections though (Sections 3.1-3.1.5); specifically regarding data sources and products used in the classifications. Therefore, we have chosen to merge Section 2.2 with Section 3.1 (Methodology), and subsequently editing to remove deprecated information. Table 1 has been kept as a simple summary of all data sources and acquisitions (and now referred to as Table 2).

Table 1: I suspect you mean 20 and 30% cloud cover limit across the whole image, not on some kind of pixel-wise basis > is that correct? It could be worth spelling this out

In the case of both Landsat 8/9 and Sentinel-2 data sources, cloud cover across a whole image is used in this filtering routine. In the Sentinel-2 imagery metadata, it is specifically referred to as a granule-specific cloudy pixel percentage for a scene footprint. This information has been added as a footnote, with reference to both cloud cover filters outlined in the table:

"a Cloud cover percentage refers to the granule-specific cloudy pixel percentage for the individual scene footprint."

L129: It is not clear to me what you mean by "where positive classifications adhere to all thresholds" > please reword or clarify

This means that a pixel is retained as water (i.e. a positive classification) only if it satisfies the threshold criteria for all five spectral indices simultaneously. This was adopted as a conservative classification approach to minimise the risk of false positives. This has been clarified in the text, updating the ambiguous statement:

"Thresholds for the indices were chosen based on previous studies of ice-marginal lakes (How et al., 2021; Shugar et al., 2020), where a pixel was classified as water only if it met the threshold criteria for all five indices." (Line 136-137)

Table 3: It would be nice to include the names of each band (e.g., "B2 = blue") in the caption for people less used to working with Sentinel data. I think maybe you are saying that it is considered water if it's less than that threshold value?

That's a good idea, so a reader does not have to go to other resources to find the band information for Sentinel-2. Band names and spatial resolutions have been added as a footnote for the table:

"aB2 = Blue (10 m); B3 = Green (10 m); B4 = Red (10 m); B8 = Near-Infrared (NIR) (10 m); B11 = Shortwave Infrared 1610 nm (SWIR1) (20 m); B12 = Shortwave Infrared 2190 nm (SWIR2) (20 m)."

L168-178: Do you have a sense how much of this variation in lake number is due to processing artifacts (e.g., one lake classified as two in some years due to data issues) as opposed to physically meaningful variations? It seems like you could do some kind of intersection/spatial join to test this.

An intersection routine is performed in post-processing on the entire inventory series to eliminate processing artefacts, such as the example you describe. Lakes are grouped consistently across all inventory years. In cases where a lake separates into two water bodies, it will still be classified as one lake across the inventory series. This is summarised in the bulletpoints in Section 3.1.4 (Inventory compilation), which summarises the post-processing curation steps. We have added the following passage to clarify that this curation includes an intersection routine to remove processing artefacts:

"- Assigning common lake identifications in instances where a lake is composed of several bodies/polygons. This is supported by automated intersection analysis, where identifications are initially defined as overlapping water bodies across all inventory years" (Line 168)

L195: It is interesting that many more lakes shrunk in size > this seems at odds with what is seen in many areas. Are you doing this analysis on a last year-first year basis? This would make your results really sensitive to noise in those years. Have you tried doing a linear fit to all lake area data at a site and making this threshold based off a rate of change rather than an absolute change? That seems like it would be more resistant to noise. Regardless, do you have any physical interpretation of why shrinking lakes are more common? Maybe this comes later.

Lake area change trends for individual lakes (presented in Figure 4) are determined on a last year-first year basis. However, we understand that this is not a robust measure of the rate of change in

lake area across the inventory series. We have therefore adopted a linear regression slope calculation in the revised manuscript to provide an improved analysis of the rate of change in lake area. This shows that 83 lakes exhibited growth between 2016 and 2023, 240 lakes experienced a decline, and 1373 lakes exhibited no significant change (i.e. ±0.05 km). This is an interesting finding, as analysis of these lake groupings show that the largest rate of area change is observed in the largest lakes in the inventory series. The majority of the "stable" lakes are the smallest lakes in the inventory series. Because of this, small and unchanging lakes account for a much smaller proportion of the total ice-marginal lake volume compared to the larger, more dynamic lakes. It is therefore likely that the regional statistics on lake area change are more influenced by shifts in the largest lakes than by trends across all regions and lake sizes. Figure 4 has been updated to include statistics from the linear regression analysis, and the associated text has been updated also:

"Overall, the rate of change in lake area could be analysed across 1696 lakes in the inventory series, representing 31% of all mapped lakes. Of these, 83 lakes showed growth between 2016 and 2023, with an increase in area of greater than 0.05 km² per year, while within 240 lakes experienced a decline, with a decrease in area of greater than 0.05 km² per year. The remaining 1,373 lakes exhibited no significant change, with area variations limited to ±0.05 km² per year (Figure 4). The largest rate of area change was observed in the larger lakes. The average area of lakes that expanded or contracted was 6.18 km² and 7.77 km², respectively, with a total combined area of 2378.75 km². In contrast, the average area of lakes that remained stable was much smaller, at 0.41 km², contributing a total combined area of 560.39 km². Therefore, while the majority of lakes experienced minimal changes in area, suggesting stability in size, they account for a much smaller proportion of the total ice-marginal lake volume compared to the larger, more dynamic lakes. It is likely that the regional statistics on lake area change are more influenced by shifts in the largest lakes than by trends across all regions and lake sizes." (Line 261-270)

In addition, a similar concern was raised in the previous round of reviews, and as a result, total (summed) lake area and median lake area are now included in Figure 3, where the total ice sheet lake area and total PGIC lake area are provided alongside lake abundance and median lake area. In addition, the associated text has been updated in Section 4.2 (Lake surface extent) describing changes in total lake area and median lake area across regions and through time (Line 235-239).

Sec 4.3 (related to Figure 6 comment): please discuss if icebergs are masked during temperature estimates. If not, it is unclear what these data mean. Also, have you analyzed lake temperature trends (or year-to-year temperature variability) on a lake-by-lake basis? This seems like it would add a lot to this section. As is, it is a little unclear what the lake surface temperature adds to the story here.

Water temperature estimates are derived from averaged scene acquisitions between July and August to minimize the likelihood of iceberg and ice cover presence. In addition, temperature estimates below 0 °C are removed to limit the influence of icebergs and ice cover. Please see the

major comments for more details, and an outline of how we have addressed this in the manuscript.

Based on previous reviewer feedback, we have now included an additional figure (Figure 7) to demonstrate lake temperature trends. Specifically, Figure 7a shows spatial trends across Greenland and highlights a latitudinal trend in lake temperatures where the northern regions (NO, NW and NE) hold lakes with cooler surface temperatures (between 0 and 4 °C) on average compared to the southern regions (SW and SE). Figure 7b shows temporal trends in lake temperature, grouped by lake size. The trend signifies a possible link between lake size and the rate of temperature change, where the smallest lakes (<= 0.5 km²) exhibited consistent surface temperatures across the inventory years whilst the largest lakes (5.0-150.0 km²) experienced the largest temperature change, cooling by an average of 0.5 °C. Lake temperature trends and Figure 7 are included in Section 4.3, following the description and discussion around Figure 6.

L229: I imagine this is described in more detail in How 2021, but it is unclear in the current manuscript how the different delineation methods are incorporated. How do you blend the datasets when they have inevitably somewhat differing shapes? In general, how do you choose whether just one or multiple methods are used to delineate a given lake?

The ice-marginal lake inventory series dataset includes all successful delineations to ensure full transparency and provide users with flexibility in how they apply the data. For the analysis of lake area presented in this study, common water bodies (i.e., those classified by more than one method) are merged based on their lake identification number to form the maximum possible extent. This merging is performed automatically and consistently using the "dissolve" function from the *geopandas* package, which unions all geometries sharing the same lake identification number and aggregates their associated attributes (Kelsey et al., 2020). This information has now been added to Section 3.1.4 (Inventory compilation), so that it is clear how the dissolve is performed:

"For the purpose of the lake abundance and area analysis presented subsequently, common water bodies (i.e. classified with more than one method) are dissolved based on lake identification number to form the maximum possible extent. Geometries with the same lake identification number are firstly merged into a single geometry using a union join, and then all geometry attributes are aggregated and combined (Kelsey et al., 2020). This is performed in an automated and consistent manner, where all successful classifications with the same identification number are dissolved in all instances." (Line 174-178)

Alongside the dataset, we provide the production pipeline which includes this routine (<a href="https://github.com/GEUS-Glaciology-and-Climate/GrIML">https://github.com/GEUS-Glaciology-and-Climate/GrIML</a>). We have now included an example tutorial of how the dissolve routine is used in the ice-marginal lake inventory series and the

expected output: <a href="https://github.com/GEUS-Glaciology-and-climate/GrIML/blob/main/tutorials/dataset tutorial.ipynb">https://github.com/GEUS-Glaciology-and-climate/GrIML/blob/main/tutorials/dataset tutorial.ipynb</a>. We have updated the Code and Data Availability section (Section 8) to highlight these tutorials to the reader:

"The production code for making the inventory series is openly available at <a href="https://github.com/GEUS-Glaciology-and-Climate/GrIML">https://github.com/GEUS-Glaciology-and-Climate/GrIML</a> (How, 2025a; b). It is distributed as a deployable and version-controlled Python package, including Jupyter Notebook tutorials on how to run the pipeline and basic handling of the dataset. If the production code is used or adapted, then we ask for a reference to be included in publications." (Line 442-445)

Sec 6.1: I think this is all true, but much applies to a time static lake inventory, so it might be useful to better articulate what having the time variation adds here.

There are many potential applications of the dataset as a static inventory; however, we have now clarified in Section 6.1 how the time-varying nature of the dataset provides additional analytical opportunities beyond those available from a single time step. Specifically, we highlight that the inventory series provides a temporal dimension that is critical for furthering the understanding of dynamic systems, such as glacial hydrology. These clarifications have been added to the beginning of Section 6.1 to better distinguish between static and time-varying uses of the dataset:

"The inventory series presented here is the first step to quantifying the terrestrial storage of meltwater, and how it changes over time, which would be highly valuable for refining estimations of the future sea level contribution of the Greenland Ice Sheet and surrounding PGICs. The dataset extends the value of a single, time-static inventory by providing a consistent, multi-year record of lake evolution. This temporal dimension enables analyses that capture lake variability and persistence through time. Tentative findings have been outlined..." (Line 354-358)

**Response to Reviewer #2**

The manuscript has been substantially improved and previous reviewer comments have been thoroughly dealt with (thanks for the detailed responses). Whilst there are some uncertainties that are not ideal, these have been minimised to a suitable level given the complexities of identifying dynamic variable targets at an almost continental scale. The multi-annual ice marginal lake inventories will be very valuable to glaciology, limnology, ecology and wider resource/hazard management across Greenland. Given the climatic variability and increased warm events these datasets will continue to gain importance and can hopefully be extended back to 2002 to give a longer time assessment. It is imperative to publish these multi-annual inventories now given the pronounced changes around Greenland and I fully support publication. I look forward to future modifications of the methodology and extensions of the time series, which will continue to push the science forward.

Thanks you for your feedback and comments on the revised manuscript. We appreciate the time and effort taken to provide this second round of comments, which we have answered. These changes largely consist of additional clarification and editing to better convey insights into the dataset.

**Minor comments**

Line 231 – 233 Slightly confusing at the moment with the term 'in general' – maybe put 'in the whole inventory' and you could add a sentence starter;

'When lake size is subcategorised by contact with the main ice sheet or PGIC then...' Maybe also worth adding a sentence explaining why this is important.

The sentence has now been changed, removing "in general" and adding a variation on the suggested opening:

"When lake surface extent is subcategorised by region then the NE region holds the largest lakes with a median lake area of 0.34 km2 (mean: 1.63 km2)." (Line 223-224)

Line 244 – Do you mean subcategorised by ice margin type? The current sentence is a bit confusing. Also add 'with subsequent time steps revealing absolute area change'?

Yes, the total summed lake area is categorised by ice margin type. This has been clarified in the text, and the suggested

"The total (summed) lake area is categorised by ice margin type (ice sheet and PGIC) in Figure 3, with subsequent time steps revealing absolute area change across each region." (Line 234-235)

Line 245 – 'through time' – I think you need to be more specific here – between 2016 and 2023. Also this is an important result in itself – I think expand on this a bit more to finish the sentence (currently feels like it's skipped over); were you expecting a uniform trend of lake area response to climate across Greenland? I think many people won't appreciate the variability in climate zones. Also possibly worth reminding the reader here that lakes losing contact with the ice margin will affect these stats – how many lakes detached in this time?

Previous studies have suggested increasing lake area through time, linked to a retreating ice margin and enhanced runoff over time (e.g. Carrivick and Quincey, 2014); however, such studies typically infer trends from discrete regions in Greenland. It is likely a uniform trend is not evident here because of the large latitudinal range and resulting climatic variability that ice-marginal lakes in Greenland are subjected to. In addition, as suggested, this could also be related to lake detachment. Currently, we cannot quantify the number of detached lakes, with lake presence in the inventory series indicating either lake detachment or methodology performance. We have added this information to the paragraph to elaborate on the possible reason for a lack of uniform trend:

"The total (summed) lake area is categorised by ice margin type (ice sheet and PGIC) in Figure 3, with subsequent time steps revealing absolute area change across each region. No inherent trends in total lake area between 2016 and 2023 are evident across all regions, such as uniform change through time. The lack of a strong trend could be related to variability in lake contact with the ice margin over time; but could also reflect the high climatic variability across Greenland given its large latitudinal range."

In addition, the two instances where the phrase "through time" are used have been changed:

"No inherent trends in total lake area between 2016 and 2023 are evident across all regions..." (Line 235-236)

"When assessing lake temperature change from 2016 to 2023, lake size appears to influence..." (Line 276)

Line 246 – add 'provides valuable'

Done.

Line 360 – Worth a comment on how the surface temperature estimates could be used? Useful for observing average conditions between years and also regional variability – importance for downstream ecology (Fellman et al., 2014).

A comment has been added to the sentence regarding the use of the inventory series in assessing surface temperature variability and its importance for downstream ecology. The citation suggested has also been used, as it is a valuable example of such a study in an Arctic region:

"Lake conditions could also provide insights into glacier dynamics in lacustrine settings around Greenland; for example, to investigate submarine melting in lacustrine settings and its impact on glacier retreat (e.g., Mallalieu et al., 2021), and to explore regional variability and evolution of lake surface temperatures and their influence on downstream ecology (e.g. Fellman et al., 2014)." (Line 359-362)

Line 380 – I think ice-marginal lake detection is far more complex than supraglacial – as this article proves! I would change 'is likely to be feasible' to 'worth further investigation'.

Done.

*Line 386 – Change 'would' to 'could'. I think 500m resolution would be too coarse.*

Done.

Line 393 – Yes SPOT 5,6 and 7 satellites would give you this and are pretty similar spectral bands to Sentinel series from what I remember. Can't you get access through ESA? I think there would be substantial changes since 2002 and would be interesting to see what impact 2012 melt season had.

The key difference with SPOT 5, 6, and 7 is that they do not include shortwave infrared (SWIR) bands. While using SPOT imagery would be beneficial because of its broader temporal coverage and higher spatial resolution, it would require us to adjust our multi-spectral indices classification method. In practice, this means we would need to simplify the approach to rely exclusively on NDWI classifications. It is likely that compromises would need to be made on the methodology to extend the inventory series back to 2002.

Figure 7 – Nice. Shows the overall regional pattern pretty well. Panel b. needs regions adding to each subplot – currently look like they're Southern regions at top and Northern regions towards the lower part of the plot – which is counterintuitive but fine if labelled.

Panel b in Figure 7 presents lake surface temperatures categorised by lake size and not regions. This has been made clearer in the figure caption:

"The temporal variability in lake surface temperature estimate is shown in (b), categorised by lake size across six size groups (up to  $0.1 \text{ km}^2$ , 0.1- $0.2 \text{ km}^2$ , 0.2- $0.5 \text{ km}^2$ , 0.5- $1.0 \text{ km}^2$ , 1.0- $1.5 \text{ km}^2$ , 1.5- $1.0 \text{ km}^2$ , and above  $1.0 \text{ km}^2$ ."

---

## Author Response (AR4)

**Response to Editor**

Thank you for your contributions to the revision of this manuscript. We greatly appreciate the constructive feedback provided by both reviewers and the thorough and thoughtful revision carried out by the authors.

From an editorial standpoint, we are pleased to accept this manuscript for publication.

One minor point for clarification: the error in lake abundance is currently shown as both a positive and negative deviation. If we interpret the authors' response correctly, this estimate reflects only a negative bias (an underestimate), not a potential positive bias (overestimate) in lake number. If so, please adjust the sign of the error in the manuscript accordingly.

We are confident that this study will be of great value to the communities investigating glaciological and hydrological processes across Greenland.

Thank you for your quick response. We are very pleased that this dataset and the associated paper meet the standards of ESSD for acceptance and publication.

We have edited the lake abundance error estimation according to your feedback and report this as a solely negative bias now (i.e. " $\pm$  809" >> "-809").

Many thanks for the valuable feedback along the way, which has ultimately strengthened and improved the dataset.

Regarding your figure 2: please note that in accordance with our standards, material that originates from Google Earth requires the copyright symbol "©". Please add it to the caption of Figure 2 with the next revision. For instance, © Google Earth Engine.

The figure does not include any content that originates from Google Earth Engine. Regardless, we have added a copyright symbol in the figure caption to ensure no copyright infringement on the use of the name in the figure.